# Robust Latent Neural Operators through Augmented Sparse Observation Encoding

## Abstract

Neural operator methods have achieved significant success in the efficient simulation and inverse problems of complex systems by learning a mapping between two infinite-dimensional Banach spaces. However, existing methods still exhibit room for optimization in terms of robustness and modeling accuracy. Specifically, existing methods are characterized by sensitivity to noise and a tendency to overlook the importance of multiple sparse observations in new domains. Therefore, we propose a robust latent neural operator based on the variational autoencoder framework. In this method, an encoder utilizing recurrent neural networks effectively captures sequential patterns and dynamical features from domain-specific sparse observations. Subsequently, a neural operator in latent space, followed by a decoder, enables the effective modeling of the original system. Additionally, for certain higher-dimensional complex systems, opting for a lower-dimensional latent space can reduce task complexity while still maintaining satisfactory modeling performance. We evaluate our approach on multiple representative systems, and experimental results demonstrate that it achieves superior modeling accuracy and enhanced robustness compared to state-of-the-art baseline methods.

## 1 Introduction

Modeling and forecasting complex dynamical systems have become central topics in scientific machine learning, with numerous effective methods developed to address these challenges Tang et al. (2020); Wang et al. (2023). These advancements are critical for unraveling complex system behaviors and facilitating downstream tasks (Li et al., 2024b; Brugere et al., 2018; Lorch et al., 2022; Li et al., 2024a; Bjørheim et al., 2022; Li et al., 2023a; Grziwotz et al., 2023). Traditional methods to dynamical system modeling primarily rely on ordinary or partial differential equations (ODEs/PDEs), but these equations exhibit limitations in more complex scenarios and are often unknown in many practical cases. To overcome these limitations, the Sparse Identification of Nonlinear Dynamics (SINDy) (Brunton et al., 2016; Kaiser et al., 2018) approach has been introduced for data-driven discovery of governing equations. More recently, leveraging the universal approximation capacity of deep neural networks (Hornik, 1991), neural network-based methods have gained increasing attention. For instance, Recurrent Neural Networks (RNNs) and their extended architectures (Memory, 2010; Cho et al., 2014; Suárez et al., 2024) have proven to be effective for processing sequential data; Neural ODEs (NODEs) (Chen et al., 2018; Li et al., 2025; Yang & Li, 2025) enable continuous-time modeling and can handle irregularly sampled observations; Graph Neural Networks (GNNs) (Murphy et al., 2021; Liu et al., 2023) provide an effective framework for systems with underlying graph structure. However, these methods typically suffer from high computational costs during inference and require retraining when environmental conditions change.

In response to the above challenge, neural operator methods have been proposed in recent years (Lu et al., 2021; Li et al., 2020; Azizzadenesheli et al., 2024; Li et al., 2024c). These methods represent an emerging class of techniques in deep learning, aimed at establishing a machine learning model capable of learning mappings between spaces of functions. Specifically, neural operators are capable of processing inputs defined by infinite-dimensional variations (such as parametric functions, initial conditions, and boundary conditions), and directly output future system states, thereby achieving modeling of a family of dynamical systems. In the context of solving PDE systems, neural operators present remarkable advantages over traditional numerical resolution methods, such as finite element and finite difference methods. These advantages include significantly higher computational speed

while maintaining comparable predictive accuracy, as well as more effective handling of diverse inverse problems (Zhao et al., 2022; Wang & Wang, 2024; Thodi et al., 2024).

However, current neural operator frameworks remain constrained by persistent challenges in terms of noise robustness and modeling accuracy. Specifically, these methods often exhibit pronounced performance degradation when handling noise-corrupted observational data, especially when the infinite-dimensional inputs of neural operators are contaminated by noise. Moreover, in new testing environments, we often observe system states at multiple moments, which are might irregularly spaced. Current approaches overlook the crucial role of these sparse observational data from new domains, particularly the sequential patterns and dynamical features embedded within them. To address this, we draw inspiration from latent flow approaches Vlachas et al. (2022); Regazzoni et al. (2024); Gao et al. (2024) and introduce a Robust Latent Neural Operator approach, termed RLNO, grounded in the variational autoencoder (VAE) framework. Specifically, the primary contributions of our work are as follows.

- A Novel Fusion Framework. RLNO represents a novel approach grounded in VAE framework, incorporating an RNN-based encoder, latent neural operator and a decoder. This framework effectively harnesses the strengths of RNNs and neural operators, enabling more accurate operator learning.

- Utilizing More Dynamic Information. The proposed OPERATOR-RNN encoder extracts and leverages more domain-specific information and dynamic evolution patterns from the sequential observational data in new domains. It remains applicable under non-uniform sampling conditions, thereby improving the robustness of the RLNO method.

- Efficiency and Scalability. RLNO inherits the low computational costs of neural operators, significantly outperforming RNN and Neural ODE-based methods. Furthermore, it can select a smaller latent space dimension, which reduces learning complexity and data requirements, enabling easier extension to high-dimensional complex systems.

## 2 RELATED WORK

Recent years have witnessed growing interest and significant progress in the field of neural operators. Among the most influential advances are the Deep Operator Network (DeepONet) introduced by Lu et al. (Lu et al., 2021; 2022), and Fourier Neural Operator (FNO) proposed by Li et al. (Li et al., 2020; Peng et al., 2024). These methods leverage the neural network model $\mathcal{G}_{\text{NN}}$ to learn the operator mapping $\mathcal{G}^{\dagger} : \mathcal{U} \to \mathcal{S}$, where $\mathcal{U}$ and $\mathcal{S}$ are two infinite-dimensional Banach spaces. Here, our research focuses on the DeepONet framework, which primarily consists of two components: the trunk network and the branch network. Specifically, the trunk network accepts an arbitrary position $y$ as input and outputs the vector $\boldsymbol{c} = \{c_1, c_2, \cdots, c_m\}$. Concurrently, the branch network takes $l$ discrete observations $\{u(x_1), u(x_2), \cdots, u(x_l)\}$ of the sampled function $u(x)$ as input and outputs the vector $\boldsymbol{b} = \{b_1, b_2, \cdots, b_m\}$. Subsequently, integrate the trunk and branch networks through the inner product of $\boldsymbol{c}$ and $\boldsymbol{b}$, thereby estimating the state value at the query point $y$. This can be represented as $\mathcal{G}_{\text{NN}}(u)(y) = \boldsymbol{c} \cdot \boldsymbol{b}$. Here, $y$ represents a position within the domain of $\mathcal{G}^{\dagger}(u)$, which could be a temporal variable or a spatial location.

Furthermore, in light of specific application scenarios, a series of enhanced neural operator methods have been successively introduced. For instance, Jin et al. proposed a multi-input neural operator based on tensor products (Jin et al., 2022), and Wang et al. introduced a physics-informed neural operator guided by known dynamical equations (Wang et al., 2021). Additionally, Cao et al. developed a Laplace neural operator capable of handling complex geometrical boundaries (Cao et al., 2024). And Kontolati et al. preliminarily validated the superiority of neural operators in the latent space (Kontolati et al., 2024). Moreover, several studies have endeavored to employ novel neural network architectures for constructing neural operators, achieving superior performance in designated tasks. This includes the use of convolutional neural networks (Raonic et al., 2023), graph neural networks (Sun et al., 2023), and Transformer structures (Hao et al., 2023; Li et al., 2023b; Shih et al., 2025), among others. These neural operator methods can effectively model a family of dynamical systems, playing a pivotal role across diverse fields such as fluid mechanics (Ye et al., 2024), material science (Oommen et al., 2024), and climate science (Jiang et al., 2023).

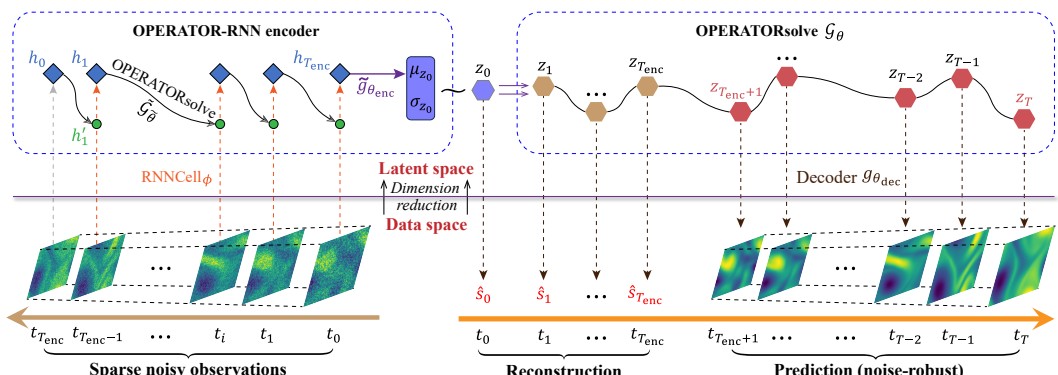

Figure 1: The sketched framework of the proposed RLNO method.

## 3 METHOD

In a standard setup, the sampling function $u$ under probability measure $\mu$ and the solution function $s$ forms $N_{\text{tr}}$ training samples $\{u^{(i)}, s^{(i)}\}_{i=1}^{N_{\text{tr}}}$. Then we train the neural operator framework RLNO with parameters $\boldsymbol{\theta}$ by minimizing $\min_{\boldsymbol{\theta} \in \Theta} \mathbb{E}_{u \sim \mu} \| \mathcal{G}^\dagger(u) - \text{RLNO}_{\boldsymbol{\theta}}(u) \|$, where $\mathcal{G}^\dagger$ denotes the true operator, $\text{RLNO}_{\boldsymbol{\theta}}$ represents our proposed neural operator, $\Theta$ is the parameter space of the constructed model. Subsequently, we elaborate on our RLNO method in detail with reference to Fig.1, with the implementation pseudocode provided in Appendix A.2.

### 3.1 NEURAL OPERATORS IN LATENT SPACE

Representations in latent spaces are ubiquitous in machine learning tasks, where the system dynamics govern the behavior of latent variables $z$ in latent space, and the system states $s$ are interpreted as manifestations of $z$ under a specific observation function $g$, which can be represented by a neural network $g_{\theta_{\text{dec}}}$ parameterized by $\theta_{\text{dec}}$. Accordingly, we need to establish a neural operator $\mathcal{G}_\theta$ parameterized by $\theta$ that maps the sampling function $u$ to the solution function $z$ in the latent space.

Subsequently, we consider the framework of VAE and employ a generative model defined by neural operators to estimate the solution function trajectory:

$$z_0 \sim p(z_0),$$
$$z_0, z_1, \cdots, z_T = \text{OPERATORsolve}(\mathcal{G}_\theta, u, z_0, (t_0, t_1, \cdots, t_T)), \quad (1)$$
$$s_i \sim p(s_i | g_{\theta_{\text{dec}}}(z_i)), \quad i = 0, 1, \cdots, T,$$

where $z_0$ signifies the initial state of the latent variable at time $t_0$ (sampled from the probability distribution $p(z_0)$), $p(z_0)$ is commonly assumed that the prior distribution of $z_0$ adheres to a Gaussian distribution, "OPERATORsolve" denotes the utilization of the neural operator $\mathcal{G}_\theta$ to determine the evolution of the latent variable, and the decoder $g_{\theta_{\text{dec}}}$ maps these latent variables to the parameters of the probability distribution $p(s_i | g_{\theta_{\text{dec}}}(z_i))$. In DeepONet framework, we initially consider a common scenario where the function $u$ corresponds to the initial value $z_0$, then we have

$$z_i = \text{OPERATORsolve}(\mathcal{G}_\theta, z_0, (t_i)) = \mathcal{G}_\theta(z_0)(t_i) = \sum_{k=1}^{m} b_k(z_0) c_k(t_i), \quad i = 0, 1, \cdots, T, \quad (2)$$

where $z$ represents the latent variable, $b_k$ is the $k$-th element of the output vector $\boldsymbol{b}$ from the branch network, $c_k$ is the $k$-th element of the output vector $\boldsymbol{c}$ from the trunk network, and $m$ represents the vector length. Similarly, for PDE systems, we have

$$z_i = \text{OPERATORsolve}(\mathcal{G}_\theta, z_0, \boldsymbol{x}, (t_i)) = \sum_{k=1}^{m} b_k(z_0) c_k(\boldsymbol{x}, t_i), \quad i = 0, 1, \cdots, T, \quad (3)$$

where $\boldsymbol{x}$ denotes spatial dimension, which takes the form $(x)$ for one-dimensional (1D) PDEs and $(x, y)$ for two-dimensional (2D) PDEs.

### 3.2 ENCODING FOR NON-UNIFORM INTERVAL OBSERVATION DATA

Inspired by the framework of Latent NODEs (Chen et al., 2018), we employ an encoder based on a reverse RNN process, which encodes sparse observations into the initial distribution of latent variables, that is, employing the posterior distribution $q(z_0|\{s_i, t_i\}_{i=0}^{T_{enc}})$ to approximate the distribution $p(z_0)$. However, in practical applications, the observational data $\{s_i, t_i\}_{i=0}^{T_{enc}}$ may not be uniformly spaced, and the traditional RNN encoder is incapable of encoding the temporal intervals. In earlier work by Rubanova et al. (Rubanova et al., 2019), two alternative approaches were introduced. One employs hidden states in RNN that decay exponentially over time, referred to as RNN-Decay encoder, and the other is based on NODE, known as the ODE-RNN encoder. However, the performance of the RNN-Decay encoder remains suboptimal, while the ODE-based process in the ODE-RNN encoder is computationally intensive, especially for high-dimensional or PDE systems.

---

**Algorithm 1** OPERATOR-RNN encoder

---

**Require:** Data and corresponding timestamps $\{s_i, t_i\}_{i=0}^{T_{enc}}$
**Ensure:** Distribution parameters for the initial state $z_0$, i.e., mean $\mu_{z_0}$ and standard deviation $\sigma_{z_0}$
1: Initialize $h_0 = 0$;
2: **for** $i$ in $1, 2, \cdots, T_{enc}$ **do**
$\quad h_i' = \text{OPERATORsolve}(\tilde{\mathcal{G}}_{\tilde{\theta}}, h_{i-1}, (t_{T_{enc}-i+1} - t_{T_{enc}-i}))$;
$\quad h_i = \text{RNNCell}_\phi(h_i', s_{T_{enc}-i})$;
$\quad$ **end for**
3: $\mu_{z_0}, \sigma_{z_0} = \tilde{g}_{\theta_{enc}}(h_{T_{enc}})$;
4: **Return:** $\mu_{z_0}, \sigma_{z_0}$

---

Therefore, we design a novel encoder based on neural operators, named OPERATOR-RNN encoder:

$$h_i' = \text{OPERATORsolve}(\tilde{\mathcal{G}}_{\tilde{\theta}}, h_{i-1}, (t_{T_{enc}-i+1} - t_{T_{enc}-i})) = \tilde{\mathcal{G}}_{\tilde{\theta}}(h_{i-1})(t_{T_{enc}-i+1} - t_{T_{enc}-i}),$$
$$h_i = \text{RNNCell}_\phi(h_i', s_{T_{enc}-i}), \qquad \text{and } i \in \{1, 2, \cdots, T_{enc}\}, \tag{4}$$

where $\tilde{\mathcal{G}}_{\tilde{\theta}}$ denotes a neural operator parameterized by $\tilde{\theta}$, and it is employed in the first equation of the OPERATOR-RNN encoder, thereby incorporating the temporal intervals into the encoder process. Finally, we map the final hidden state $h_{T_{enc}}$ to the mean $\mu_{z_0}$ and standard deviation $\sigma_{z_0}$ of the distribution $q(z_0|\{s_i, t_i\}_{i=0}^{T_{enc}})$ through a three-layer neural network $\tilde{g}_{\theta_{enc}}$ parameterized by $\theta_{enc}$, i.e., $\mu_{z_0}, \sigma_{z_0} = \tilde{g}_{\theta_{enc}}(h_{T_{dec}})$. The pseudocode for the execution process of the OPERATOR-RNN encoder is provided in Algorithm 1, succinctly described as

$$q(z_0|\{s_i, t_i\}_{i=0}^{T_{enc}}) = \mathcal{N}(\mu_{z_0}, \sigma_{z_0}),$$
$$\mu_{z_0}, \sigma_{z_0} = \text{OPERATOR-RNN}_{\theta_{enc}, \tilde{\theta}, \phi}(\{s_i, t_i\}_{i=0}^{T_{enc}}), \tag{5}$$

where $\theta_{enc}$, $\tilde{\theta}$, and $\phi$ represent the trainable parameters of the OPERATOR-RNN encoder. It should be noted that in the above process, we feed the observational data in reverse order from $t_{T_{enc}}$ to $t_0$.

### 3.3 TRAINING RLNO USING EVIDENCE LOWER BOUND

Finally, we train the encoder, decoder, and the neural operator in latent space concurrently by maximizing the Evidence Lower Bound (ELBO):

$$\text{ELBO}(\theta_{enc}, \tilde{\theta}, \phi, \theta, \theta_{dec}) = \mathbb{E}_{z_0 \sim q_{\theta_{enc}, \tilde{\theta}, \phi}(z_0|\{s_i, t_i\}_{i=0}^{T_{enc}})} \left[\log p_{\theta, \theta_{dec}}(s_0, s_1, \cdots, s_T|z_0)\right]$$
$$- \text{KL}\left(q_{\theta_{enc}, \tilde{\theta}, \phi}\left(z_0|\{s_i, t_i\}_{i=0}^{T_{enc}}\right) \| p(z_0)\right), \tag{6}$$

where the first term represents the data likelihood, the second term denotes the Kullback-Leibler (KL) divergence between the prior distribution $p$ and the estimated distribution $q$ of the initial state $z_0$, $\boldsymbol{\theta} = (\theta_{enc}, \tilde{\theta}, \phi, \theta, \theta_{dec})$ are trainable parameters, and the detailed derivation process is provided in Appendix A.1. Notably, the first term comprises $T$ data points, while the OPERATOR-RNN encoder considers only the initial $T_{enc}$ data points. Given the reality of sparse observations and the necessity of extrapolation prediction, we typically adopt $T \gg T_{enc}$. As a result, the trained RLNO model is highly effective for both system reconstruction and prediction tasks.

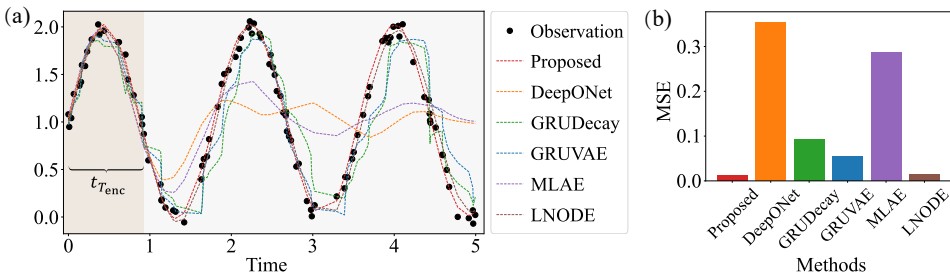

Figure 2: The experimental results of toy dataset using different methods. (a) The prediction performance in a test data. (b) The mean squared error (MSE) of predictions by various methods.

### 3.4 THE CASE OF MULTI-INPUT FUNCTIONS

Beyond the initial hidden state $z_0$, our approach may incorporate additional sampling functions as inputs, such as parameter functions and boundary conditions. Without loss of generality, we consider an additional sampling function $u$. Following the approach outlined in (Jin et al., 2022), we can extend the RLNO method. Specifically, we introduce a new branch network and incorporate the additional sampling function $u$ as input, and it outputs the vector $\{d_1, d_2, \cdots, d_m\}$. Consequently, the neural operator $\mathcal{G}_\theta$ depicted in Eq. (2) can be extended to:

$$z_i = \text{OPERATORsolve}(\mathcal{G}_\theta, z_0, u, (t_i)) = \mathcal{G}_\theta(z_0)(u)(t_i)$$
$$= \sum_{k=1}^{m} b_k(z_0) d_k(u) c_k(t_i), \quad i = 0, 1, \cdots, T. \tag{7}$$

For clarity, we refer to the aforementioned extension method as Multi Input RLNO (MI-RLNO).

## 4 EXPERIMENTS

In this section, we conduct experiments on a server equipped with 64GB RAM and an NVIDIA RTX 4090 GPU with 25.2GB memory, and provide a detailed analysis of our method across several representative systems. In addition, we introduce the Gaussian observational noise with a mean of 0 and a standard deviation of $\sigma_n$ into the experiment data. To validate the effectiveness of our method, we benchmark our experimental results against the state-of-the-art (SOTA) baseline methods, namely: DeepONet (Lu et al., 2021), Multi-Input DeepONet (MI-DON) (Jin et al., 2022), GRU Variational Autoencoder (GRUVAE) (Rubanova et al., 2019), GRU Decay (GRUDecay) (Che et al., 2018), Latent DeepONet with a multi-layer autoencoder (MLAE) (Kontolati et al., 2024), Latent Neural ODE (LNODE) (Rubanova et al., 2019), PDE-Net (Long et al., 2018) and FNO (Li et al., 2020). And the implementation details and discussion about the above baseline methods can be found in Appendix B. Additionally, we detail the hyperparameter settings of our RLNO method in Appendix C.

### 4.1 TOY DATASET

First, we consider a toy dataset of periodic trajectories from the LNODE reference (Rubanova et al., 2019). Specifically, we take $\sigma_n = 0.2$ and $T = 100$ to generat 2000 trajectories with non-equidistant observations, allocating 80% for training and 20% for testing. The frequency and initial value of each trajectory are randomly sampled from the uniform distributions. Then we set $T_{enc} = 10$ for recognition network and employ various methods to learn the dynamics of these trajectories, and the experimental results are shown in Fig. 2. It is evident from Fig. 2 that our RLNO method effectively predicts this family of systems, exhibiting the lowest prediction error compared to baseline methods.

Under the conditions of unknown frequencies and high noise, our RLNO method significantly outperforms traditional baseline methods based on RNNs and neural operators, thereby preliminarily demonstrating its advantages in terms of noise robustness and effective utilization of sparse observational data. Notably, in this process, our OPERATOR-RNN encoder can effectively extract frequency information from the first $t_{T_{enc}}$ sparse observations. In addition, although the LNODE method exhibits comparable predictive performance to our method in this toy experiment, it suffers from catastrophic failure in terms of computational cost and training convergence as dimensionality increases, particularly for the PDE systems discussed subsequently.

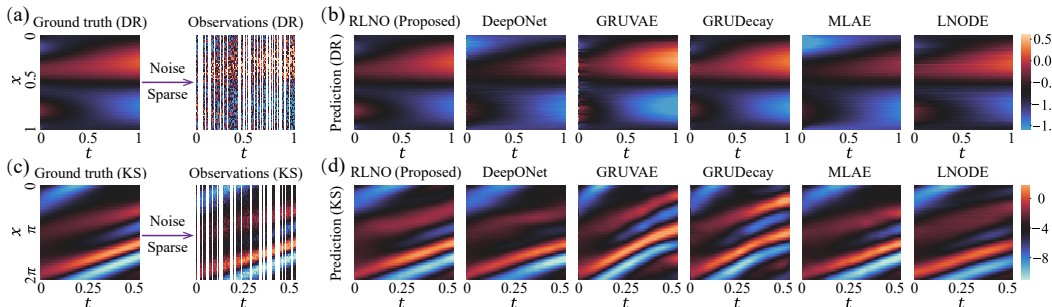

Figure 3: Experimental results of various methods in DR and KS systems. (a), (c) True and observed values in the DR and KS systems. (b), (d) Prediction results in the DR and KS systems.

## 4.2 1D PDE Systems

We then validate our approach on two representative 1D PDE systems, including the Diffusion-Reaction (DR) equation (Eq. (8)) and the Kuramoto-Sivashinsky (KS) equation (Eq. (9)) systems:

$$\partial_t s = 0.01\partial_{xx}s + 0.01s^2 + u_1(x,t), \ x \in [0,1], \ t \in [0,1], \tag{8}$$

$$\partial_t s = -\partial_x(s^2/2) - \partial_{xx}s - u_2(x,t)\partial_{xxxx}s, \ x \in [0,2\pi], \ t \in [0,0.5], \tag{9}$$

where $u_1$ and $u_2$ represent the parameter functions. To validate our method, we construct a family of dynamical systems and performing experiments via two strategies. The first involves generating initial values $s_0$ using Gaussian random fields (GRFs) with a radial-basis function kernel, given by

$$s_0 \sim c \cdot G\left\{\mu_G, \exp\left[-||x_1 - x_2||^2/(2l^2)\right]\right\}, \tag{10}$$

where $\mu_G$ represents the mean value, $x_1$ and $x_2$ are two points within the domain, $l$ is the length scale governing the smoothness, and $c$ is the scaling factor for the output. The second strategy pertains to the random generation of parameters $u$ through GRFs. Additionally, each piece of data comprises $T$ equidistant data points, and then sparse observational data is created by randomly retaining a proportion $\lambda$ of these points. Unless otherwise specified, we default the parameter $T_{\text{enc}}$ to 10.

For case 1, we fix the parameters $u_1 \equiv 0$ and $u_2 \equiv 0.081$, and employ GRFs to generate the initial condition $s_0$, thereby modeling a family of dynamical systems with varying initial values. In DR experiments, experimental data were generated based on the following parameter settings: GRF parameters $c = 1.0$, $l = 0.2$, the numbers of training and test samples $N_{\text{tr}} = 8000$, $N_{\text{te}} = 2000$, the number of sampling points and step size $T = 100$, $\Delta t = 0.01$, sparsity parameter $\lambda = 0.6$, and noise intensity $\sigma_{\text{n}} = 0.4$. Figure 3(a) randomly presents a sample of test data. After training, the predictive performances of RLNO and the baseline methods are depicted in Fig. 3(b), with the corresponding Mean Squared Error (MSE) results reported in the first row of Table 1. In KS experiments, we generate experimental data with the following parameters: $c = 10$, $l = 0.2$, $N_{\text{tr}} = 4000$, $N_{\text{te}} = 1000$, $T = 100$, $\Delta t = 0.005$, $\lambda = 0.6$, $\sigma_{\text{n}} = 1.0$, and an example of test data is shown in Fig. 3(c). After training, the predictive performance is illustrated in Fig.3(d), with the corresponding MSE results detailed in the second row of Table 1. Despite the chaotic and complex behavior exhibited by the KS system, our approach maintains a more robust and accurate predictive performance, significantly surpassing other baseline methods. In this context, even when the LNODE method employs smaller simulation steps and higher-precision numerical solvers, it fails to achieve predictive performance comparable to that of the RLNO method.

To further validate the effectiveness of our approach, two additional experiments are considered. Initially, for the DR system, we fix the initial condition $s_0$ and utilize the GRF to generate the parameter function $u$, thereby modeling a family of dynamical systems with varying parameters. Here, we take $\sigma_{\text{n}} = 0.1$ and maintain all other experimental parameters consistent with those of the DR (case 1). As illustrated in the third row of Table 1, it is evident that our method achieves the lowest prediction error. It is noteworthy that in the experiments of DR (case 2), we configure $u(x)$ as a time-invariant sampling function. In this context, our approach can accurately model the neural operator even without knowledge of the sampling function $u(x)$. This is attributed to the RLNO method's capacity to capture the underlying dynamics related to $u(x)$ via non-uniform sparse observations. In

Table 1: Comparing the MSE ($\pm$ two standard deviations) across multiple experiments

| Systems | (MI-) DON | GRUVAE | GRUDecay | MLAE | LNODE | FNO | (MI-) RLNO |
|---|---|---|---|---|---|---|---|
| DR (case 1) | $0.0114_{\pm 0.0146}$ | $0.0255_{\pm 0.0279}$ | $0.0181_{\pm 0.0214}$ | $0.0094_{\pm 0.0119}$ | $0.0033_{\pm 0.0045}$ | $0.0238_{\pm 0.0245}$ | $\mathbf{0.0012}_{\pm \mathbf{0.0016}}$ |
| KS (case 1) | $0.3697_{\pm 0.8492}$ | $3.1764_{\pm 5.1487}$ | $2.8613_{\pm 4.5272}$ | $0.4116_{\pm 0.5953}$ | $0.3212_{\pm 0.7671}$ | $0.4438_{\pm 0.7112}$ | $\mathbf{0.1230}_{\pm \mathbf{0.2325}}$ |
| DR (case 2) | $0.2240_{\pm 0.3976}$ | $0.0080_{\pm 0.0140}$ | $0.0216_{\pm 0.0479}$ | $0.1632_{\pm 0.1888}$ | $0.0219_{\pm 0.0475}$ | $0.2921_{\pm 0.3727}$ | $\mathbf{0.0017}_{\pm \mathbf{0.0020}}$ |
| KS (case 2) | $0.5686_{\pm 0.8873}$ | $5.0330_{\pm 9.1339}$ | $4.9334_{\pm 8.7576}$ | $0.8876_{\pm 1.2794}$ | $0.9740_{\pm 1.3793}$ | $0.7478_{\pm 0.8183}$ | $\mathbf{0.2389}_{\pm \mathbf{0.3637}}$ |
| NS (case 1) | $0.0013_{\pm 0.0009}$ | $0.0030_{\pm 0.0054}$ | $0.0052_{\pm 0.0048}$ | $0.0037_{\pm 0.0055}$ | $0.0015_{\pm 0.0017}$ | $0.0024_{\pm 0.0009}$ | $\mathbf{0.0005}_{\pm \mathbf{0.0004}}$ |
| NS (case 2) | $0.0009_{\pm 0.0017}$ | $0.0291_{\pm 0.0306}$ | $0.0417_{\pm 0.0437}$ | $0.0032_{\pm 0.0041}$ | $0.0026_{\pm 0.0108}$ | $0.0040_{\pm 0.0064}$ | $\mathbf{0.0001}_{\pm \mathbf{0.0002}}$ |

such scenarios, RNN-based methods (GRUVAE and GRUDecay) outperform methods solely based on neural operators (DeepONet, FNO and MLAE). Our RLNO method, which leverages the advantages of RNN encoding and neural operators, achieves optimal performance.

Finally, we examine a more complex scenario involving the KS system where both the initial condition $s_0$ and the parameter function $u$ are randomly generated by GRFs. Here, $u(t)$ is a time-varying sampling function with GRF parameters $c = 0.01$ and $l = 0.5$, and all other parameters are consistent with those used in the experiments of KS (case 1). In this scenario, we employ the MI-RLNO method to incorporate $u(t)$ as an additional input, and utilize MI-DON as the first baseline method. As demonstrated in the fourth row of Table 1, it is evident that our approach yields the most superior predictive performance. Additionally, methods based on neural operators outperform those based on RNNs, attributed to the utilization of $u(t)$ information. Moreover, we also apply the PDE-NET approach to the PDE systems, but it struggles to accurately infer the underlying equations in noisy environments, causing predictive errors to escalate rapidly and leading to divergence. Thus, this paper does not specifically present the predictive results of the PDE-NET baseline method.

### 4.3 2D PDE System

Next, we extend the application of the proposed method to 2D PDE systems. We first consider the 2D Navier-Stokes (NS) equations for a viscous, in compressible fluid in vorticity form, which reads

$$\partial_t s = \partial_x \gamma \partial_y s - \partial_y \gamma \partial_x s + 0.001 \Delta s + u(x, y, t), \ \Delta \gamma = -s, \quad (11)$$

where $(x, y) \in [0, 2]^2$, $t \in [0, 3]$, $\gamma$ is the stream function, $\Delta$ represents the Laplacian operator. Subsequently, we generate experimental data based on the following experimental parameters: $N_{tr} = 2000$, $N_{te} = 500$, $T = 100$, $\Delta t = 0.03$, $\gamma = 0.6$, $\sigma_n = 0.2$. Similarly, here we conduct experiments under two distinct scenarios. For case 1, we consider a spatial resolution of $32 \times 32$ and employ a 2D GRFs to randomly generate the initial condition $s_0$, thereby modeling a family of systems with varying initial conditions. After training, the experimental results of our method are illustrated in Fig. 4(a), with the predictive MSE of various methods presented in the fifth row of Table 1. For case 2, we consider a spatial resolution of $64 \times 64$, and employ a 2D GRFs to randomly generate initial values $s_0$ and parameters $u$. The experimental outcomes are presented in Fig. 4(b) and the last row of Table 1. The results from these experiments indicate that our proposed RLNO method is capable of utilizing sparse observations to achieve more robust and accurate modeling.

Moreover, we conduct experiments in the scenario of Rayleigh–Bénard (RB) convection, with the parameters set to $\lambda = 0.9$, $T_{enc} = 6$, $\sigma_n = 0.1$, and all other experimental setup consistent with the latest literature (Kontolati et al., 2024) (see also Appendix E). The experimental results demonstrate that our RLNO approach (MSE $= 0.0038 \pm 0.0036$) achieves a 35.6% reduction in MSE compared to the SOTA method MLAE (MSE $= 0.0059 \pm 0.0051$), with the results of one test case illustrated in Fig 4(c). Therefore, our method exhibits superior performance in more complex 2D PDE scenarios.

### 4.4 Ablation Experiments and Robustness Analysis

In this section, we validate the robustness of the RLNO method under varying parameter settings through experiments and demonstrate the advantages of the OPERATOR-RNN encoder and the VAE framework through ablation studies. Here, we design the following four ablation experiments. "Ab1": we modify the RLNO framework by substituting the OPERATOR-RNN encoder with a standard RNN encoder; "Ab2": within the RLNO framework, we replace the OPERATOR-RNN

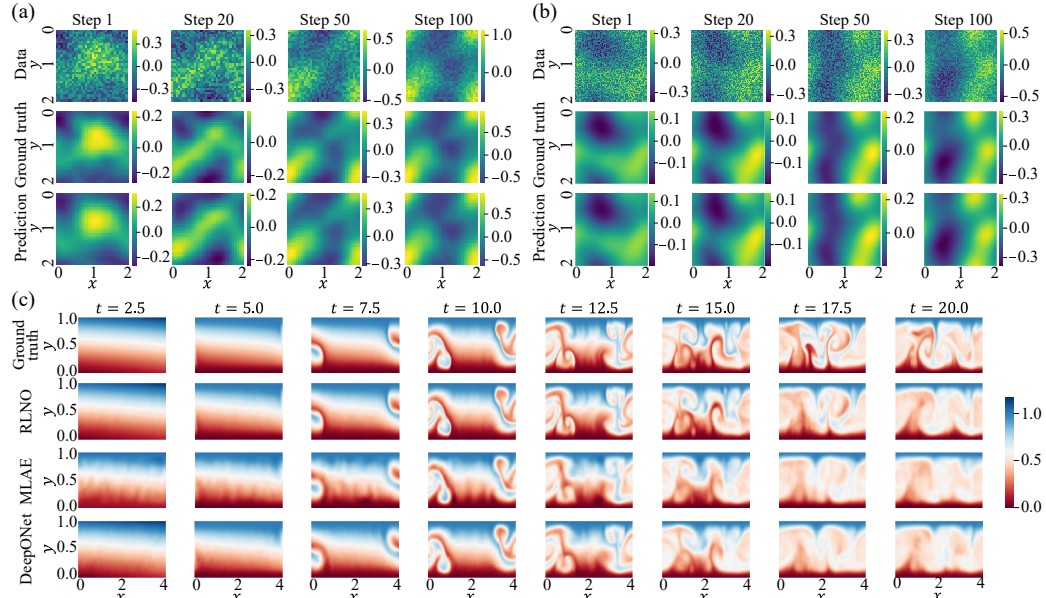

Figure 4: Experimental performance in NS and RB systems using RLNO method. (a) Experimental results for NS (case 1). (b) Experimental results for NS (case 2). (c) Experimental performance in RB system using different methods.

encoder with the RNN-Decay encoder; "Ab3": we replace the OPERATOR-RNN encoder with the ODE-RNN encoder; "Ab4": we alter the loss function from the ELBO to MSE.

We begin by evaluating the performance of DR (Case 1) under varying noise levels and training samples. The prediction results are presented in Figs. 5(a) - 5(b), Table S2 and Figs. S1–S2 in Appendix D. Here, the sparsity parameter is fixed at $\lambda = 0.2$; Fig. 5(a) corresponds to $N_{tr} = 4000$, while Fig. 5(b) uses $\sigma_n = 1.0$, with all other parameters held constant. Experimental results indicate that under low-noise conditions, all methods achieve commendable performance, with RLNO and Ab2 exhibiting optimal outcomes. As noise levels increase, the superiority of the RLNO method becomes more pronounced. On one hand, our method demonstrates lower prediction errors compared to Ab1–Ab3, confirming the superiority of the OPERATOR-RNN encoder; on the other hand, Ab4 exhibits the highest prediction error, supporting the claim that training with the ELBO indeed enhances noise robustness. Moreover, our method consistently exhibits superior performance across different training samples $N_{tr}$, with the advantage becoming particularly pronounced when abundant data is available. This indicates that the OPERATOR-RNN encoder, combined with the ELBO training objective, effectively harnesses the data to capture more accurate underlying dynamics.

Additionally, we conduct experiments of KS (case 1) under the condition of varying recognition lengths ($T_{enc}$), and keep all other experimental parameters consistent with previous settings. The predicted MSE is presented in Fig. 5(c), Table S3 and Fig. S3 in Appendix D. The experimental results demonstrate that an increase in $T_{enc}$ enhances the amount of dynamic information encoded in the initial values of the latent variables, thereby improving model performance. Moreover, increasing $T_{enc}$ leads to a diminishing marginal return in modeling performance. This phenomenon can be attributed to two principal factors: first, the fixed number of RNN neurons entails an upper limit on its capacity for encoding information; second, the intrinsic memory decay attribute of RNNs leads to a gradual forgetting of inputs from the more distant past. Nevertheless, incorporating sparse observational data can substantially enhance the predictive capability of neural operators.

Finally, we opt for varying latent space dimensions $d_z$ across four experiments: DR (case 1), KS (case 1), NS (case 1), and NS (case 2). The experimental outcomes (see Table 2) indicate that despite the relatively high dimensionality of the experimental data spaces (64 dimensions for the DR and KS experiments, $32 \times 32$ dimensions for NS (case 1), and $64 \times 64$ dimensions for NS (case 2)), it is feasible to use lower-dimensional latent spaces while still maintaining high modeling accuracy. Our experiments partially reveal the features redundant in the original data, suggesting that this dimensionality reduction can decrease the complexity of the tasks at hand. However, the latent

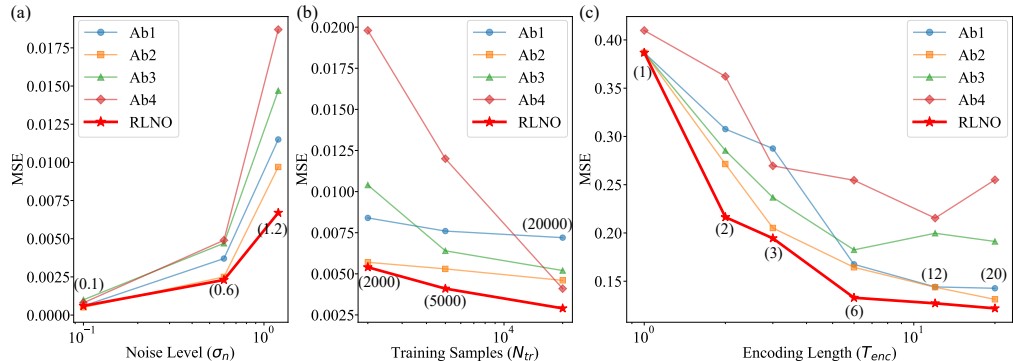

Figure 5: Comparison of prediction MSE in ablation studies. (a) Ablation study under different noise levels; (b) Ablation study with varying training sample sizes; (c) Ablation study with different encoding lengths. Values in parentheses are the corresponding abscissas for each set of results.

Table 2: Comparing the MSE ($\pm$ two standard deviations) across different latent space dimensions

| Systems | $d_z = 8$ | $d_z = 16$ | $d_z = 32$ | $d_z = 64$ | $d_z = 128$ | $d_z = 256$ |
|---|---|---|---|---|---|---|
| DR (case 1) | 0.0025 $\pm 0.0089$ | 0.0022 $\pm 0.0072$ | 0.0021 $\pm 0.0063$ | 0.0021 $\pm 0.0049$ | 0.0021 $\pm 0.0073$ | 0.0020 $\pm 0.0067$ |
| KS (case 1) | 0.4991 $\pm 0.8232$ | 0.1473 $\pm 0.3068$ | 0.1390 $\pm 0.2893$ | 0.1334 $\pm 0.2742$ | 0.1304 $\pm 0.2518$ | 0.1368 $\pm 0.2625$ |
| NS (case 1) | 0.0024 $\pm 0.0027$ | 0.0019 $\pm 0.0020$ | 0.0013 $\pm 0.0007$ | 0.0012 $\pm 0.0007$ | 0.0013 $\pm 0.0007$ | 0.0011 $\pm 0.0006$ |
| NS (case 2) | 4.6e-4 $\pm 3.6\text{e-}4$ | 2.8e-4 $\pm 2.0\text{e-}4$ | 2.1e-4 $\pm 1.6\text{e-}4$ | 1.5e-4 $\pm 1.3\text{e-}4$ | 1.4e-4 $\pm 1.1\text{e-}4$ | 1.6e-4 $\pm 1.6\text{e-}4$ |

space should not be overly compressed, as the optimal size is contingent upon the complexity of the task itself. For instance, in the more complex experiment of NS (Case 2), modeling effectiveness noticeably improves as $d_z$ increases up to 64 (as shown in the Fig. S4 and Table 2), indicating $d_z = 64$ is effective despite being much smaller than the original $64 \times 64$ space. These findings indicate that our approach is capable of extracting critical dynamical information within a family of systems, thereby facilitating its application to higher-dimensional and more complex PDE systems.

## 5 CONCLUDING REMARKS

This work introduces RLNO, a novel neural operator grounded in the framework of VAEs. This method initially encodes the original system states into a latent space. then predicts the future states via a neural operator $\mathcal{G}_\theta$ in latent space. Finally, a decoder maps the predicted latent variables back to the data space, enabling more accurate operator learning. In this process, our model is trained by maximizing the evidence lower bound, and the Gaussian prior assumption within the model enables our method to more effectively handle observations with noise. We conduct experiments across several parametric ODE and PDE systems, and the results demonstrate that our RLNO method surpasses SOTA baseline methods in terms of noise robustness and modeling accuracy.

To enhance the sparse observation encoding in new domains, our RLNO method incorporates an OPERATOR-RNN encoder. This encoder inputs multiple observational data and encodes temporal interval and dynamical patterns of sequential data, thereby exhibiting superior modeling performance over traditional RNN-based and NODE-based encoders in irregularly spaced observations, which can be validated through meticulously designed ablation studies. Additionally, we conduct a parameter sensitivity analysis on the dimensionality of latent space. Experimental results indicate that lower-dimensional setting in latent space achieve impressive modeling accuracy. This dimensionality reduction from data space to latent space decreases the task's complexity, facilitating the neural operator's ability to capture essential features with limited training data. Consequently, our method can be readily extended to model higher-dimensional complex systems. However, current work has limitations that warrant further investigation. For example, this paper primarily focuses on operator learning tasks under simple sampling distributions (e.g., GRFs), while future work should extend the experimental analysis to more complex and realistic scenarios. This is of significant practical importance for replacing inefficient numerical simulations with efficient neural operators.

## REPRODUCIBILITY STATEMENT

To ensure the reproducibility of our work, we have provided the complete source code and instructions necessary to replicate all experimental results reported in this paper. The code, along with details on dependencies and hyperparameter settings, is included in our supplementary materials.

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

# Appendix

*Notes: We use the large language models solely for the purpose of language polishing.*

## A   THE EXECUTION DETAILS OF THE RLNO METHOD

### A.1   APPROXIMATE POSTERIOR AND EVIDENCE LOWER BOUND

In this section, we provide a detailed description of the approximate posterior distribution $q$ mentioned in the main text, and derive the optimization objective, the Evidence Lower Bound (ELBO).

Given the data $s_{0:T} = \{s_0, s_1, \cdots, s_T\}$ from the original system, our primary interest in variational inference (VI) tasks lies in the posterior distribution of the latent variables, $p(z_0|s_{0:T})$. Evaluating this distribution directly is often challenging or impractical due to the involved marginalization operations, which require integration or summation over all possible configurations of the latent variables—a task that becomes exceedingly complex in high-dimensional spaces. VI addresses this issue by introducing a more tractable distribution, $q(z_0)$, as an approximation of the posterior distribution. Due to

$$\log p(s_{0:T}) = \log \int p(s_{0:T}, z_0) \mathrm{d}z_0 = \log \int p(s_{0:T}, z_0) \frac{q(z_0)}{q(z_0)} \mathrm{d}z_0$$
$$= \log \mathbb{E}_{q(z_0)} \left[ \frac{p(s_{0:T}, z_0)}{q(z_0)} \right] \geq \mathbb{E}_{q(z_0)} \left[ \log \frac{p(s_{0:T}, z_0)}{q(z_0)} \right] = \mathrm{ELBO}, \tag{S1}$$

the ELBO serves as a lower bound for the log evidence, also known as the log marginal likelihood. Through derivation, we can obtain:

$$\mathrm{ELBO} = \int q(z_0) \log \frac{p(s_{0:T}, z_0)}{q(z_0)} \mathrm{d}z_0 = \int q(z_0) \log \frac{p(s_{0:T}|z_0)p(z_0)}{q(z_0)} \mathrm{d}z_0$$
$$= \int q(z_0) \log p(s_{0:T}|z_0) \mathrm{d}z_0 - \int q(z_0) \log \frac{q(z_0)}{p(z_0)} \mathrm{d}z_0 \tag{S2}$$
$$= \mathcal{L}_1 - \mathcal{L}_2.$$

In accordance with Equation (6) in the main text, it is evident that the posterior distribution $q(z_0)$ is determined by the observational data $s_{0:T_{\mathrm{enc}}}$ within the recognition network. This relationship can succinctly be expressed as follows:

$$z_0 \sim q_{\theta_{\mathrm{enc}}, \tilde{\theta}, \phi}(z_0 | \{s_i, t_i\}_{i=0}^{T_{\mathrm{enc}}}), \tag{S3}$$

where $\theta_{\mathrm{enc}}, \tilde{\theta}$, and $\phi$ represent the trainable parameters in the recognition network as defined in the main text. Therefore, we have:

$$\mathcal{L}_1 = \mathbb{E}_{z_0 \sim q_{\theta_{\mathrm{enc}}, \tilde{\theta}, \phi}(z_0 | \{s_i, t_i\}_{i=0}^{T_{\mathrm{enc}}})}[\log p_{\theta, \theta_{\mathrm{dec}}}(s_0, s_1, \cdots, s_T | z_0)], \tag{S4}$$

where $\theta$ and $\theta_{\mathrm{dec}}$ respectively represent the trainable parameters of the neural operator $\mathcal{G}_\theta$ in the latent space and the decoder $g_{\theta_{\mathrm{dec}}}$. And $\mathcal{L}_2$ refers to the Kullback-Leibler (KL) divergence:

$$\mathcal{L}_2 = \mathrm{KL}\left( q_{\theta_{\mathrm{enc}}, \tilde{\theta}, \phi}\left( z_0 | \{s_i, t_i\}_{i=0}^{T_{\mathrm{enc}}} \right) \| p(z_0) \right). \tag{S5}$$

Through the aforementioned process, we transform the optimization problem of maximizing $\log p(s_{0:T})$ into maximizing its evidence lower bound.

### A.2   THE EXECUTION PSEUDOCODE OF THE RLNO METHOD

To more effectively illustrate the implementation details of the proposed RLNO method, this section presents its pseudocode in Algorithm 2 and Algorithm 3. Before presenting the algorithm pseudocode, the key notation used throughout this paper are summarized in Table S1. These notations can be better understood in conjunction with Fig. 1 of the main text.

In testing process, this methodology takes as input the observed data from a new domain, $\{s_i, t_i\}_{i=0}^{T_{\mathrm{enc}}}$, along with a sampling function $u$ from the domain, and outputs direct predictions of the system states

Table S1: Key Notation

| Notation | Description |
|---|---|
| $s$ | system variable in data space |
| $z$ | system variable in latent space |
| $u$ | the input function, such as initial/boundary conditions or parametric functions |
| $\tilde{g}_{\theta_{\text{enc}}}$ | an encoder parameterized by $\theta_{\text{enc}}$ |
| $\text{RNNCell}_\phi$ | the backward RNN cell within the encoder, parameterized by $\phi$ |
| $\tilde{\mathcal{G}}_{\tilde{\theta}}$ | the neural operator within the encoder, parameterized by $\tilde{\theta}$ |
| $\mathcal{G}_\theta$ | the neural operator in latent space, parameterized by $\theta$ |
| $g_{\theta_{\text{dec}}}$ | a decoder parameterized by $\theta_{\text{dec}}$ |
| $\mu_{z_0}, \sigma_{z_0}$ | the mean and variance of the initial value $z_0$, which are estimated by the encoder |
| $T_{\text{enc}}$ | the encoding length, namely the number of sparse observations used by the encoder |
| $T$ | the total number of prediction steps, typically set such that $T \gg T_{\text{enc}}$. |
| $N_{\text{tr}}$ | the size of the training set |
| $\sigma_{\text{n}}$ | the standard deviation of Gaussian noise, reflecting the noise level |
| $d$ | the dimensionality of the data space |
| $d_z$ | the dimensionality of the latent space |

---

**Algorithm 2** The Training Process of the RLNO Method

---

**Require:** Given $N_{\text{tr}}$ training data instances, each instance comprises $T$ observed states $\{s_i, t_i\}_{i=0}^T$, along with the corresponding sampling function $u$

**Ensure:** The trained parameters $\theta, \tilde{\theta}, \phi, \theta_{\text{enc}}, \theta_{\text{dec}}$

1: Initialize $h_0 = 0$;
2: **for** $i$ in $1, 2, \cdots, T_{\text{enc}}$ **do**
    $h_i' = \text{OPERATORsolve}(\tilde{\mathcal{G}}_{\tilde{\theta}}, h_{i-1}, (t_{T_{\text{enc}}-i+1} - t_{T_{\text{enc}}-i}))$;
    $h_i = \text{RNNCell}_\phi(h_i', s_{T_{\text{enc}}-i})$;
    **end for**
3: $\mu_{z_0}, \sigma_{z_0} = \tilde{g}_{\theta_{\text{enc}}}(h_{T_{\text{enc}}})$;
4: Sampling $z_0$ from a Gaussian distribution with mean $\mu_{z_0}$ and variance $\sigma_{z_0}^2$;
5: $z_0, z_1, \ldots, z_T = \text{OPERATORsolve}(\mathcal{G}_\theta, u, z_0, (t_0, t_1, \ldots, t_T))$;
6: $\hat{s}_0, \hat{s}_1, \ldots, \hat{s}_T = g_{\theta_{\text{dec}}}(z_0, z_1, \ldots, z_T)$;
7: Use Equation (7) in the main text to calculate ELBO, and use it as the loss function to train parameters $\theta, \tilde{\theta}, \phi, \theta_{\text{enc}}, \theta_{\text{dec}}$;
8: **Return**: After sufficient training, output the trained parameters $\theta, \tilde{\theta}, \phi, \theta_{\text{enc}}, \theta_{\text{dec}}$

---

**Algorithm 3** The Testing Process of the RLNO Method

---

**Require:** Sparse observations in a new domain $\{s_i, t_i\}_{i=0}^{T_{\text{enc}}}$, sampling function $u$

**Ensure:** Prediction of the system state, $s(\tau)$, for $\tau \in [t_0, t_T]$, where $T_{\text{enc}} \ll T$

1: Initialize $h_0 = 0$;
2: **for** $i$ in $1, 2, \cdots, T_{\text{enc}}$ **do**
    $h_i' = \text{OPERATORsolve}(\tilde{\mathcal{G}}_{\tilde{\theta}}, h_{i-1}, (t_{T_{\text{enc}}-i+1} - t_{T_{\text{enc}}-i}))$;
    $h_i = \text{RNNCell}_\phi(h_i', s_{T_{\text{enc}}-i})$;
    **end for**
3: $\mu_{z_0}, \sigma_{z_0} = \tilde{g}_{\theta_{\text{enc}}}(h_{T_{\text{enc}}})$;
4: Sampling $z_0$ from a Gaussian distribution with mean $\mu_{z_0}$ and variance $\sigma_{z_0}^2$;
5: $z_\tau = \text{OPERATORsolve}(\mathcal{G}_\theta, u, z_0, (\tau)), \tau \in [t_0, t_T]$;
6: $s_\tau = g_{\theta_{\text{dec}}}(z_\tau), \tau \in [t_0, t_T]$;
7: **Return**: $s_\tau, \tau \in [t_0, t_T]$

---

$s_\tau$, $\tau \in [t_0, t_T]$. It is crucial to note that the length of the sparse observations $T_{\text{enc}}$ is typically set much shorter than the total data length $T$, i.e., $T_{\text{enc}} \ll T$.

In addition, in our experiments, the term OPERATORsolve denotes the execution of operator computations utilizing the DeepONet framework, RNNCell employs the GRU module, and the symbols $\tilde{g}$ and $g$ denote the usage of a three-layer feedforward neural network, respectively.

## A.3 Universal Approximation Theorem for RLNO

To theoretically validate the effectiveness of the proposed method, we analogously present the Universal Approximation Theorem for RLNO, as seen in Theorem 1. Subsequently, we briefly provide the proof of this theorem by integrating the proof process of the classic DeepONet method.

**Theorem 1.** *(**Universal Approximation Theorem for RLNO**). Suppose that $\mathcal{X}$ is a Banach space, $K_1 \subset \mathcal{X}$, $K_2 \subset \mathbb{R}^d$ are two compact sets in $\mathcal{X}$ and $\mathbb{R}^d$, respectively, $V$ is a compact set in $C(K_1)$. Assume that $\mathcal{G}$ is a nonlinear continuous operator, which maps $V$ into $C(K_2)$. Then for any $\epsilon > 0$, there are positive integers $l$, $m$, continuous vector functions $\boldsymbol{f} : \mathbb{R}^{l+d_z} \to \mathbb{R}^m$, $\tilde{\boldsymbol{f}} : \mathbb{R}^d \to \mathbb{R}^m$, and $x_1, x_2, \ldots, x_m \in K_1$, such that*

$$\left| \mathcal{G}(u)(y) - g_{dec}\left( \langle \boldsymbol{f}\left(u(x_1), u(x_2), \cdots, u(x_l), z_0\right), \tilde{\boldsymbol{f}}(y) \rangle \right) \right| < \epsilon \tag{S6}$$

*holds for all $u \in V$ and $y \in K_2$, where $\langle \cdot, \cdot \rangle$ denotes the dot product in $\mathbb{R}^m$, and*

$$z_0 \sim q(z_0 | \{s_i, t_i\}_{i=0}^{T_{enc}}) = \mathcal{N}(\mu_{z_0}, \sigma_{z_0}),$$
$$\mu_{z_0}, \sigma_{z_0} = \text{OPERATOR-RNN}(\{s_i, t_i\}_{i=0}^{T_{enc}}), \tag{S7}$$

*where OPERATOR-RNN serves as an encoder for sparse observations $\{s_i, t_i\}$ from a specific new domain in $V$.*

*Proof.* In fact, when the conditions

$$g_{\text{dec}}(z) = z,$$
$$\boldsymbol{f}\left(u(x_1), u(x_2), \cdots, u(x_l), z_0\right) = \boldsymbol{f}\left(u(x_1), u(x_2), \cdots, u(x_l)\right) \tag{S8}$$

are met, Theorem 1 simplifies to Theorem 2 as presented in Reference (Lu et al., 2021). Therefore, by fixing Equation (S8) and based on the proof in Reference (Lu et al., 2021), it is established that there exist neural networks $\boldsymbol{f}$ and $\tilde{\boldsymbol{f}}$ such that Theorem 1 holds. $\qquad \square$

In practical applications, both the OPERATOR-RNN and $g_{\text{dec}}$ are typically trainable neural networks serving as the encoder and decoder, respectively. Generally, the degenerate case represented by Equation (S8) is not considered. Additional encoder and decoder can significantly enhance the robustness and accuracy of neural operators, as can be substantiated by the following three considerations.

- First, the VAE framework transforms the operator modeling issue in the original data space into an operator modeling problem in the latent space, offering enhanced flexibility and generality. For example, traditional DeepONet approaches, which model the dynamics of observed variables directly, may struggle to capture the implicit relationships within high-dimensional data. In contrast, our method, by compressing or expanding observed data into a latent representation, then modeling the dynamical system within this latent space, is more adept at capturing complex temporal sequence characteristics.

- Second, our approach employs variational inference to model latent variables, enhancing its suitability for handling complex data characterized by uncertainty or noise. In fact, ablation studies and comparative experiments with classic baseline methods convincingly support this conclusion.

- Third, an RNN-based encoder (OPERATOR-RNN) can efficiently leverage additional observed states and potential dynamics information in new domains. In contrast, classical neural operator approaches fail to achieve this. In fact, this can be attributed to the fact that the initial value vectors of latent variables can generate system states under specific sampling functions more robustly and accurately through neural operators in the latent space, thereby modeling a family of dynamical systems.

## B  INTRODUCTION AND DISCUSSION OF BASELINE METHODS

In this section, we offer a succinct overview of the deployment of baseline methodologies within our experiments.

First, we consider the original version of the Deep Operator Network method (Lu et al., 2021), herein referred to as DeepONet. This technique is structured around a branch network and a trunk network. The branch network processes $l$ equidistantly distributed sampling points from the parameter function $u(\boldsymbol{x}, t)$, generating a $m$-dimensional vector $\boldsymbol{b} = \{b_1, \ldots, b_m\}$. Conversely, the trunk network accepts the temporal and spatial variables, $t$ and $\boldsymbol{x}$, respectively, yielding another $m$-dimensional vector $\boldsymbol{c} = \{c_1, \ldots, c_m\}$. Consequently, the prediction of the system's state for any given set of inputs is articulated as follows:

$$\mathscr{G}(u)(\boldsymbol{x}, t) = \sum_{k=1}^{m} b_k(u) c_k(\boldsymbol{x}, t) + q,$$

where $q \in \mathbb{R}$ represents a bias term. After training, we can accurately forecast the state value $s(\boldsymbol{x}, t)$ for any sampled function $u$. Additionally, when both the initial value $s_0$ and parameter $u$ vary, we employ the MI-DON approach (Jin et al., 2022). Specifically, we consider introducing a new branch network that takes the initial value $s_0$ as its input and outputs an $m$-dimensional vector $\boldsymbol{d} = \{d_1, \cdots, d_m\}$, thereby facilitating the prediction of the system's future states:

$$\mathscr{G}(s_0)(u)(\boldsymbol{x}, t) = \sum_{k=1}^{m} b_k(u) c_k(\boldsymbol{x}, t) d_k(s_0) + q,$$

where $q$ is a bias term.

Second, we consider GRU Variational Autoencoder (GRUVAE) method (Rubanova et al., 2019), which is fundamentally an RNN prediction method in latent space. Specifically, the approach initially employs an encoder to map $s_0$ to the initial value $z_0$ in the latent space. Subsequently, it predicts the future states of latent variables through the GRU framework, given by:

$$h_i, z_i = \text{GRUCell}_\phi(h_{i-1}, z_{i-1}), \quad \text{and } i \in \{1, 2, \cdots, T\}. \tag{S9}$$

where $\text{GURCell}_\phi$ is GRU module parameterized by $\phi$. And ultimately, decodes the latent variables back to their original states.

Third, we examine the GRU method with an exponential decay in time (Che et al., 2018), henceforth referred to as GRUDecay. The method described shares similarities with the second approach in that it is also based on RNN. However, the distinctive aspect of GRUDecay lies in its consideration of temporal interval information. Specifically, the GRU state update equation in the hidden space is formulated as follows:

$$\begin{aligned} h_i, z_i &= \text{GRUCell}_\phi(h_{i-1}, t_i - t_{i-1}, z_{i-1}) \\ &= \text{GRUCell}_\phi(h_{i-1} \cdot \exp\{-\tau \Delta_t(i)\}, z_{i-1}), \end{aligned} \tag{S10}$$

where $i \in \{1, 2, \cdots, T\}$, $\Delta_t(i) = t_i - t_{i-1}$ represents the time interval.

Fourth, we consider Latent DeepONet with a multi-layer autoencoder (MLAE) (Kontolati et al., 2024). This method is a two-stage training approach within a latent space. In the first stage, the authors pre-train an autoencoder using a multilayer fully connected neural network. This autoencoder is capable of encoding raw spatial data into a latent space. Subsequently, a DeepONet method is trained within this latent space to predict the state of latent variables at any given time. There are two fundamental distinctions between the MLAE method and our RLNO approach. Firstly, our method's OPERATOR-RNN encoder capitalizes on multiple sparse observational data inputs. Secondly, we employ the VAE framework to train all parameters concurrently.

Fifth, we consider Latent Neural Ordinary Differential Equation (Latent NODE) (Rubanova et al., 2019). This approach shares similarities with our RLNO method, with the primary distinctions being two-fold: Firstly, the method utilizes an ODE-RNN encoder for its recognition network; secondly, it employs "ODEsolve" for predicting the states of latent variables. Despite latent NODE method exhibiting strong predictive performance in numerous low-dimensional systems, it encounters unacceptably high computational complexity when applied to large-scale, high-dimensional systems or PDE systems.

Table S2: Experimental hyperparameters in different systems

| Experiment | $N_{\text{tr}}$ | $N_{\text{te}}$ | $N_x$ | $T$ | $T_{\text{enc}}$ | $\Delta t$ | $l$ | $c$ | $\lambda$ | $\sigma_{\text{n}}$ | $d_z$ | $d_{\text{rec}}$ |
|---|---|---|---|---|---|---|---|---|---|---|---|---|
| Toy model | 1600 | 400 | / | 100 | 10 | / | / | / | / | 0.2 | 32 | 32 |
| DR (case 1) | 8000 | 2000 | 64 | 100 | 10 | 0.01 | 0.2 | 1.0 | 0.6 | 0.4 | 64 | 64 |
| DR (case 2) | 8000 | 2000 | 64 | 100 | 10 | 0.01 | 0.2 | 1.0 | 0.6 | 0.1 | 64 | 64 |
| KS (case 1) | 4000 | 1000 | 64 | 100 | 10 | 0.005 | 0.2 | 10 | 0.6 | 1.0 | 64 | 64 |
| KS (case 2) | 4000 | 1000 | 64 | 100 | 10 | 0.005 | 0.5 | 0.01 | 0.6 | 1.0 | 64 | 64 |
| NS (case 1) | 4000 | 1000 | $32 \times 32$ | 100 | 10 | 0.03 | / | / | 0.6 | 0.2 | 64 | 100 |
| NS (case 2) | 4000 | 1000 | $64 \times 64$ | 100 | 10 | 0.03 | / | / | 0.6 | 0.2 | 64 | 100 |
| RB | 720 | 80 | $128 \times 128$ | 50 | 5 | 0.5 | / | / | 0.9 | 0.1 | 100 | 100 |

Sixth, we consider the PDE-Net approach (Long et al., 2018), which leverages finite differences to approximate spatial derivative terms and uses simple backward Euler for training and testing. In particular, for 2-d PDE systems, this method employs specific convolution kernels to compute spatial derivatives. Experiments reveal that the method underperforms when it fails to accurately infer the underlying dynamical equations, inevitably leading to significant prediction errors in roll-out forecasts.

Seventh, we consider the Fourier Neural Operator (FNO) approach (Li et al., 2020), which formulates a neural operator by directly parameterizing the integral kernel in Fourier space. In practice, we take the system state $s(\boldsymbol{x}, t)$ and the parameter function $u(\boldsymbol{x}, t)$ at time $t$ as input and directly output the system state $s(\boldsymbol{x}, t + \delta_t)$ at time $t + \delta_t$. In fact, in our experimental setup, this method is unable to achieve super-resolution prediction along the temporal dimension. Consequently, we opt for predicting the system state at sampling points $\{t_0, t_0 + \Delta t, \cdots, t_0 + T\Delta t\}$ with $\delta_t = \Delta t$.

## C    EXPERIMENTAL PARAMETERS SETTINGS

To facilitate the replication of our experiments, in this section, we provide a detailed account of the hyperparameter settings used in the main text. Here, $N_{\text{tr}}$ denotes the number of training sets, $N_{\text{te}}$ represents the number of test sets, $N_x$ indicates the spatial discretization dimensionality of PDEs, $T$ stands for the number of sampled data points, $T_{\text{enc}}$ signifies the length of the encoder network, $\Delta t$ refers to the sampling time step, $l$ and $c$ respectively correspond to the length scale and scaling factor of GRF, $\lambda$ represents the sparsification ratio, $d_z$ denotes the dimensionality of the latent space for the neural operator, and $d_{\text{rec}}$ represents the dimensionality of the latent space for the recognition network.

## D    SUPPLEMENTARY EXPERIMENTAL RESULTS

### D.1    SUPPLEMENTARY RESULTS IN ABLATION ANALYSIS

In this section, we supplement the analysis results of ablation studies mentioned in the main text, refer to Section 4.4. First, we compared the prediction errors of different methods under varying noise levels $\sigma_{\text{n}}$ and training samples $N_{\text{tr}}$, as summarized in Table S3. This was followed by an evaluation of their performance across different encoding lengths $T_{\text{enc}}$, detailed in Table S4. As shown in Fig. 5 of the main text and Tables S3 – S4, the proposed RLNO method achieves not only smaller prediction errors but also lower standard deviations, demonstrating superior statistical stability and enhanced predictive performance.

### D.2    SUPPLEMENTARY RESULTS IN ROBUSTNESS ANALYSIS

In this section, we supplement the robustness analysis results mentioned in the main text. For detailed information, please refer to Section 4.4 of the main text and Figures S1-S6.

Table S3: Comparing the MSE ($\pm$ two standard deviations) across different $\sigma_n$ and $N_{tr}$

| Methods | Noise Level ($\sigma_n$) | | | Training Samples ($N_{tr}$) | | |
|---|---|---|---|---|---|---|
| | 0.1 | 0.6 | 1.2 | 2000 | 5000 | 20000 |
| Ab1 | $0.0006_{\pm 0.0025}$ | $0.0037_{\pm 0.0068}$ | $0.0115_{\pm 0.0146}$ | $0.0084_{\pm 0.0243}$ | $0.0076_{\pm 0.0107}$ | $0.0072_{\pm 0.0070}$ |
| Ab2 | $\mathbf{0.0005}_{\pm \mathbf{0.0028}}$ | $0.0025_{\pm 0.0066}$ | $0.0097_{\pm 0.0134}$ | $\mathbf{0.0057}_{\pm \mathbf{0.0179}}$ | $0.0053_{\pm 0.0093}$ | $0.0046_{\pm 0.0051}$ |
| Ab3 | $0.0010_{\pm 0.0038}$ | $0.0047_{\pm 0.0103}$ | $0.0147_{\pm 0.0212}$ | $0.0104_{\pm 0.0219}$ | $0.0064_{\pm 0.0138}$ | $0.0052_{\pm 0.0050}$ |
| Ab4 | $0.0008_{\pm 0.0043}$ | $0.0049_{\pm 0.0186}$ | $0.0187_{\pm 0.0324}$ | $0.0198_{\pm 0.0488}$ | $0.0120_{\pm 0.0271}$ | $0.0041_{\pm 0.0046}$ |
| RLNO | $\mathbf{0.0006}_{\pm \mathbf{0.0025}}$ | $\mathbf{0.0023}_{\pm \mathbf{0.0048}}$ | $\mathbf{0.0067}_{\pm \mathbf{0.0114}}$ | $\mathbf{0.0054}_{\pm \mathbf{0.0183}}$ | $\mathbf{0.0041}_{\pm \mathbf{0.0091}}$ | $\mathbf{0.0029}_{\pm \mathbf{0.0038}}$ |

Table S4: Comparing the MSE ($\pm$ two standard deviations) across different encoding lengths $T_{enc}$

| Methods | $T_{enc} = 1$ | $T_{enc} = 2$ | $T_{enc} = 3$ | $T_{enc} = 6$ | $T_{enc} = 12$ | $T_{enc} = 20$ |
|---|---|---|---|---|---|---|
| Ab1 | $0.3868_{\pm 0.6011}$ | $0.3076_{\pm 0.4414}$ | $0.2875_{\pm 0.4163}$ | $0.1675_{\pm 0.3338}$ | $0.1441_{\pm 0.3047}$ | $0.1427_{\pm 0.3327}$ |
| Ab2 | $0.3868_{\pm 0.6011}$ | $0.2714_{\pm 0.4295}$ | $0.2052_{\pm 0.3524}$ | $0.1644_{\pm 0.2949}$ | $0.1438_{\pm 0.2879}$ | $0.1312_{\pm 0.2712}$ |
| Ab3 | $0.3868_{\pm 0.6011}$ | $0.2854_{\pm 0.3218}$ | $0.2368_{\pm 0.4255}$ | $0.1825_{\pm 0.4707}$ | $0.1997_{\pm 0.4132}$ | $0.1912_{\pm 0.6320}$ |
| Ab4 | $0.4098_{\pm 0.9593}$ | $0.3622_{\pm 0.5661}$ | $0.2695_{\pm 0.5768}$ | $0.2546_{\pm 0.3707}$ | $0.2154_{\pm 0.3624}$ | $0.2551_{\pm 0.5151}$ |
| RLNO | $0.3868_{\pm 0.6011}$ | $\mathbf{0.2164}_{\pm \mathbf{0.3872}}$ | $\mathbf{0.1945}_{\pm \mathbf{0.3287}}$ | $\mathbf{0.1329}_{\pm \mathbf{0.3023}}$ | $\mathbf{0.1271}_{\pm \mathbf{0.2456}}$ | $\mathbf{0.1219}_{\pm \mathbf{0.2165}}$ |

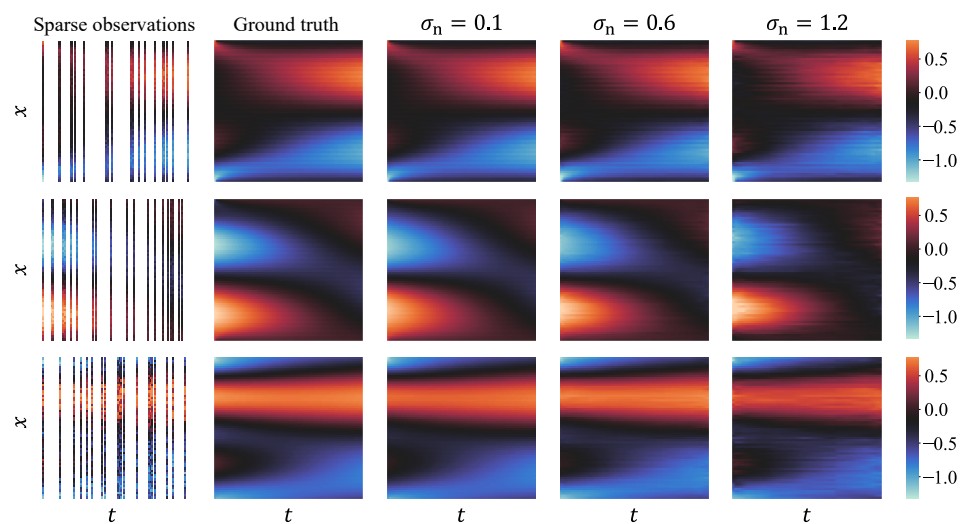

Figure S1: Experimental results across different $\sigma_n$ in three test data of DR (case 1) experiment.

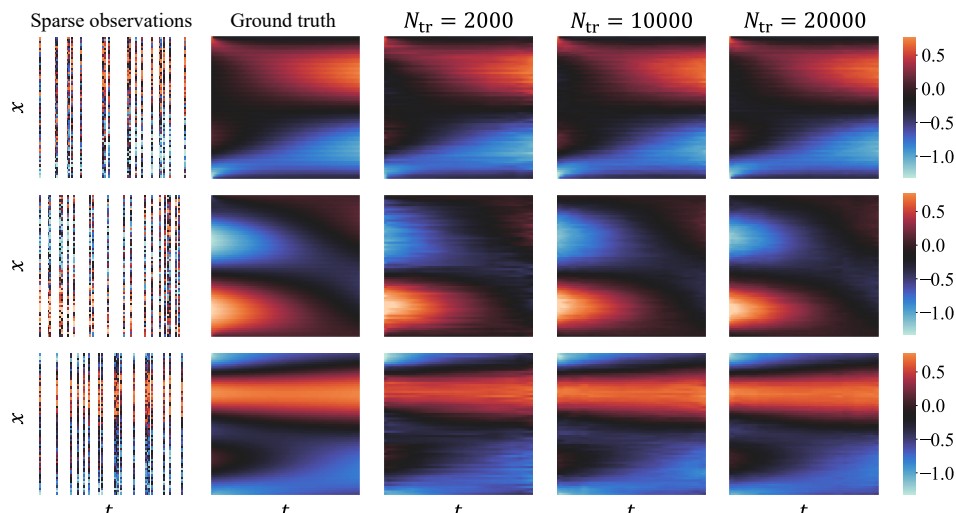

Figure S2: Experimental results across different $N_{\mathrm{tr}}$ in three test data of DR (case 1) experiment.

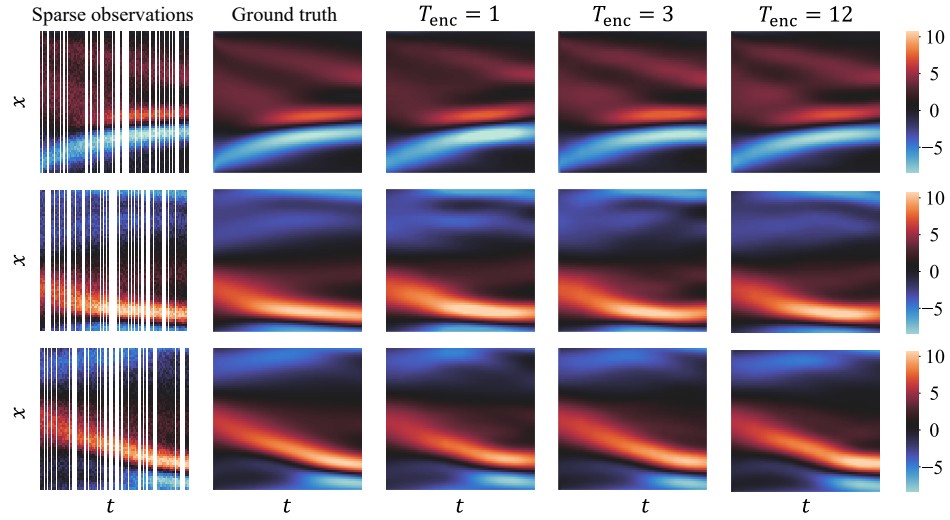

Figure S3: Experimental results across different $T_{\mathrm{enc}}$ in three test data of KS (case 1) experiment.

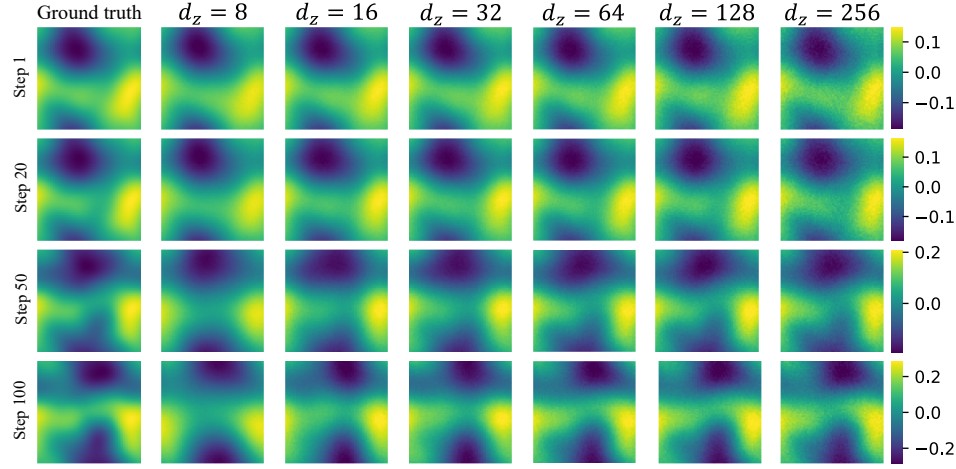

Figure S4: Experimental results across different $d_z$ in a test data of NS (case 2) experiment.

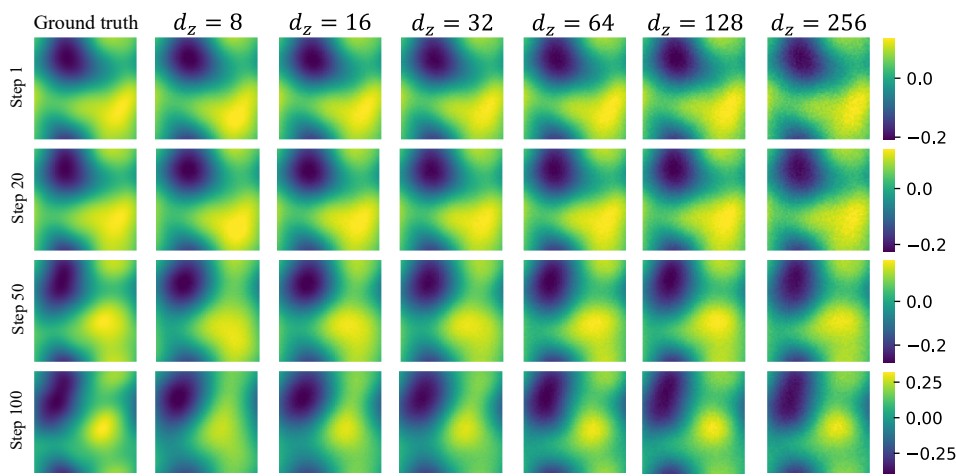

Figure S5: Experimental results across different $d_z$ in a test data of NS (case 2) experiment.

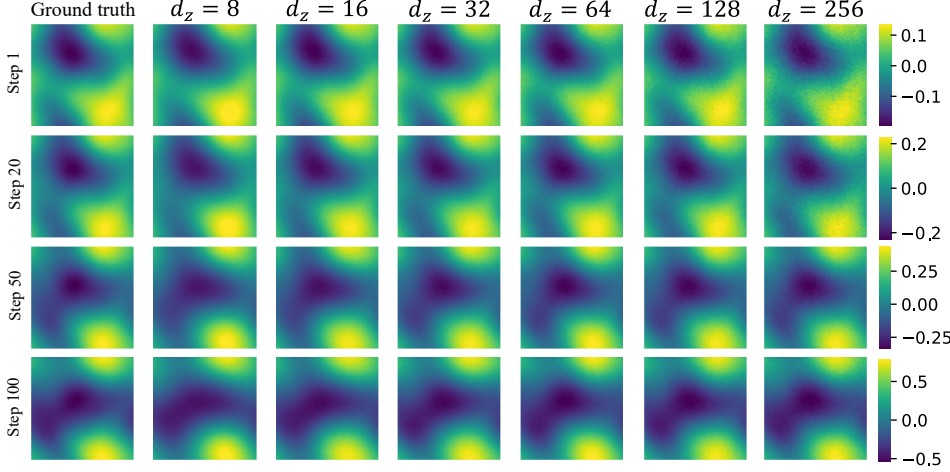

Figure S6: Experimental results across different $d_z$ in a test data of NS (case 2) experiment.

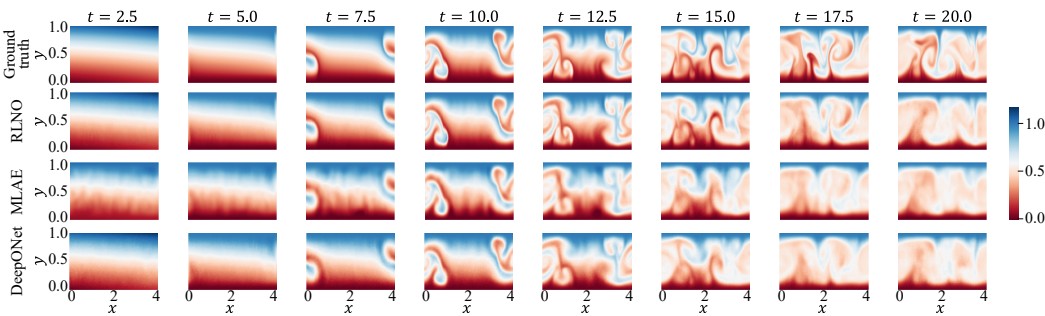

Figure S7: Experimental performance in RB system using different methods.

# E    RAYLEIGH-BÉNARD FLUID FLOW CONVECTION

Rayleigh-Bénard convection arises in a thin fluid layer subjected to basal heating. This natural fluid convection is driven by buoyancy due to a temperature gradient $\Delta T$. When $\Delta T$ is large enough that the Rayleigh number Ra exceeds a critical threshold, flow instability emerges.

Next, we generate experimental data following the description in (Kontolati et al., 2024). The dimensionless Rayleigh-Bénard system under the Boussinesq approximation (mass, momentum, energy conservation for incompressible flow in $\Omega$) is:

$$\begin{cases} \frac{D\boldsymbol{u}}{Dt} = -\frac{1}{\rho_0}\nabla p + \frac{\rho}{\rho_0}g + \nu\nabla^2\boldsymbol{u}, & \boldsymbol{x} \in \Omega, t > 0, \\ \frac{DT}{Dt} = \kappa\nabla^2 T, & \boldsymbol{x} \in \Omega, t > 0, \\ \nabla \cdot \boldsymbol{u} = 0, & \\ \rho = \rho_0(1 - \alpha(T - T_0)), & \end{cases}$$

where $\alpha$ is the thermal expansion coefficient, $g$ is the gravitational acceleration, $\nu$ is the kinematic viscosity, $\kappa$ is the thermal diffusivity, $D/Dt$ denotes the material derivative, while $\boldsymbol{u}$, $p$, and $T$ represent the fluid velocity, pressure, and temperature, respectively. $T_0$ is the temperature at the lower plate, and $\boldsymbol{x} = (x, y)$ denotes spatial coordinates. Boundary conditions (BCs) and initial conditions (ICs) are defined for the upper and lower plates:

$$\begin{cases} T(\boldsymbol{x}, t)|_{y=0} = T_0, & \boldsymbol{x} \in \Omega, t > 0, \\ T(\boldsymbol{x}, t)|_{y=h} = T_1, & \boldsymbol{x} \in \Omega, t > 0, \\ \boldsymbol{u}(\boldsymbol{x}, t)|_{y=0} = \boldsymbol{u}(\boldsymbol{x}, t)|_{y=h} = 0, & \boldsymbol{x} \in \Omega, t > 0, \\ T(y, t)|_{t=0} = T_0 + \frac{y}{h}(T_1 - T_0) + 0.1\nu(\boldsymbol{x}), & \boldsymbol{x} \in \Omega, \\ \boldsymbol{u}(\boldsymbol{x}, t)|_{t=0} = 0, & \boldsymbol{x} \in \Omega, \end{cases}$$

where $h$ is the thickness of the fluid layer, $T_0$, and $T_1$ are the fixed temperatures of the lower and upper plates, respectively. We aim to approximate the operator $\mathcal{G} : T(\boldsymbol{x}, t = 0) \mapsto T(\boldsymbol{x}, t)$, which maps the initial temperature field to its time evolution. The simulation is conducted on a spherical domain $\Omega = [0, 4] \times [0, 1]$, discretized into a $128 \times 128$ spatial grid. For each realization, the PDE is solved over $t \in [0, 20]$ with a timestep $\Delta t = 10^{-2}$. The dimensionless Rayleigh number is set to $2 \cdot 10^6$ and the Prandtl number to 1. We generate $N = 800$ samples, split into $N_{\text{train}} = 720$ for training and $N_{\text{test}} = 80$ for testing. The dataset is generated using the Dedalus Project (https://github.com/DedalusProject/dedalus).

To complement the evaluation of prediction performance on the test data in the main text, we present here a comparative results of different methods on a training sample, as illustrated in Fig. S7. Our experimental results show that although all compared methods demonstrate competitive performance, the proposed RLNO method achieves significantly higher precision in detailed predictions while maintaining the lowest mean squared error (MSE) among all approaches.

