# OpenReview forum: "Robust Latent Neural Operators through Augmented Sparse Observation Encoding"
_ICLR.cc/2026/Conference — ICLR 2026 Conference Withdrawn Submission_

### Official Review · Reviewer_KSwG · 2025-10-30

**Soundness:** 3
**Presentation:** 3
**Contribution:** 3
**Rating:** 8
**Confidence:** 2

**Summary:**

The authors propose a model called RLNO with VAE framework for operator learning. The proposed model has a RNN-based encoder, neura operator in the latent space, and decoder. This framework enables us to deal with noisy and sparse input data with low computational costs.

**Strengths:**

The proposed model is noise robust, computationally efficient, and can deal with temporal time interval inputs. They empirically show the performance of the proposed model with various situations.

**Weaknesses:**

The presentation should be improved. For example, please define $b_k$ and $c_k$ in Eq. (2) and what is $x_1$ and $x_2$ in Eq. (10)? Also, in line 289, equation  (Eq. equation 8) shoud be equation (8).

**Questions:**

- In Section 3.2, the authors insist that an advantage of the proposed model over the ODE-RNN method is the computational cost. Comparing the computational time and the MSE of the proposed model and the ODE-RNN model would be interesting.
- The authors insist that the proposed model accepts time series with nonconstant time intervals. Does the performance changes between the constant and nonconstant time interval cases?

---

> ### Author Response · Authors · 2025-11-16
> **To Reviewer KSwG’s Report**
>
> We would like to thank the reviewer for the positive feedback and helpful suggestions. We also kindly recommend that the reviewer reads the "**General response**" for a comprehensive overview of our key contributions and the main revisions.
>
> ```
> Weakness: The presentation should be improved. For example, please define $b_k$ and $c_k$ in Eq. (2) and what is $x_1$ and $x_2$ in Eq. (10)? Also, in line 289, equation (Eq. equation 8) shoud be equation (8).
> ```
> **Response**: Thank you for your careful reading and helpful suggestions. Regarding the definitions of $b_k$ and $c_k$ in Eq. (2), they represent the $k$-th elements of the output vectors from the branch network and trunk network, respectively. This was previously introduced in the first paragraph of Section 2. As for $x_1$ and $x_2$ in Eq. (10), they refer to two points within the domain of the Gaussian Random Field (GRF). In the revised version, we have placed the relevant explanations after their respective equations for better clarity. In addition, the typos in line 289 have been corrected to “Eq. (8)” and “Eq. (9)”.
>
> Finally, we perform a careful final pass over the manuscript to refine and supplement relevant technical details, thereby enhancing both the rigor and readability of the paper.
>
>
> ```
> Question 1: In Section 3.2, the authors insist that an advantage of the proposed model over the ODE-RNN method is the computational cost. Comparing the computational time and the MSE of the proposed model and the ODE-RNN model would be interesting.
> ```
> **Response**: Many thanks for your constructive feedback. We agree that a direct comparison of computational time and MSE would be valuable. In fact, we have compared our method with the  LNODE method (with ODE-RNN encoder) on the periodic orbits toy example. As shown in Figure 2 of the main text, although LNODE achieves near-optimal prediction accuracy, its computational time increases significantly, with an average runtime 1–2 orders of magnitude higher than the proposed RLNO method.
>
> Moreover, this computational cost gap becomes drastically more pronounced in more complex PDE systems, rendering LNODE’s execution time prohibitive and often causing it to rapidly diverge or fail to execute. Therefore, we present a qualitative description instead of a precise timing comparison in the main text.
>
>
> ```
> Question 2: The authors insist that the proposed model accepts time series with nonconstant time intervals. Does the performance changes between the constant and nonconstant time interval cases?
> ```
> **Response**: Many thanks for your insightful comments. In fact, the ability to handle time series with non-uniform time intervals represents a key advantage of the proposed method. In real-world scenarios, sparsely and irregularly sampled observational data are frequently encountered, which often pose greater challenges for machine learning tasks compared to uniformly sampled data.  This is because in the case of non-uniform sampling, we need to encode not only the system states but also additional temporal interval information, which significantly increases the task complexity. In contrast, for uniformly sampled data, there is no need to account for temporal intervals (as they remain constant). Consequently, when applied to uniformly sampled time series, our method generally achieves better predictive performance.
>
> In simulation experiments of this study, the data is first generated using equally spaced sampling. Subsequently, we control the sparsity level by introducing a parameter $\lambda \in (0,1]$, which determines the proportion of data retained (i.e., randomly discarding $1-\lambda$ data points). We find that a larger $\lambda$ makes it easier to capture the underlying patterns in the input data, thereby facilitating more accurate learning of the true dynamics in new domain system.
>
>
> Finally, we hope that our **"General response"** as well as the individual responses adequately address your concerns. For your convenience, the responses above are provided in a concise and direct manner. Should you have any further inquiries or require additional clarification, we look forward to your response and welcome the opportunity to engage in a more detailed discussion.

---

### Official Review · Reviewer_GZq4 · 2025-10-30

**Soundness:** 3
**Presentation:** 3
**Contribution:** 2
**Rating:** 4
**Confidence:** 3

**Summary:**

The paper proposes the Robust Latent Neural Operator (RLNO), a variational autoencoder–like latent neural operator framework. The authors introduce an encoder, termed the OPERATOR-RNN, which takes as input a short window of possibly irregularly sampled time series data and outputs a Gaussian posterior over a finite-dimensional latent vector  $𝑧_0$. A latent DeepONet is then used as the evolution map in the latent space, generating a trajectory that is decoded back to the observed state space. The training objective maximizes an ELBO loss over $𝑧_0$. The authors perform ablations  are consider where the encoder is substituted with baselines, and vary the encoder window and the latent vector size. The authors consider a series of experiments on simple PDEs and compare against baselines with favorable results.

**Strengths:**

- The encoder architecture is a reasonable engineering upgrade to ODE/RNN/Decay approaches as it targets irregular sampling specifically.
- The ablations are useful as they show how choices affect the prediction capabilities of the model. Also, they consider noisy signals.
- The methods seems to be more accurate than the competition in the chosen metric.
- Moreover, the encoder architecture represents a reasonable engineering refinement over existing ODE-, RNN-, or Decay-based approaches, as it is specifically designed to handle irregularly sampled data. However, methods for tackling non-equidistant sampling have already been explored in the literature, including graph-based and recurrent–convolutional hybrids such as the GNN-tCNN and LSTM-tCNN architectures, which also operate on irregularly spaced observations.

**Weaknesses:**

- The main novelty of the paper is the encoder engineering improvement. Different latent flow approaches such that Learning Effective Dynamics (see “Multiscale simulations of complex systems by learning their effective dynamics”), Latent ODEs (Learning the intrinsic dynamics of spatio-temporal processes through Latent Dynamics Networks), and generative diffusion models (see Generative learning for forecasting the dynamics of high-dimensional complex systems) which the authors do not compare against. What the authors propose is in a way a subset of these methods.
- DeepONet is only effective when the outputs of the neural operator lie in a linear span of trunk features, which is not the case for complex physical systems. I would suggest the author to consider the approach described in Non linear Manifold Decoders for Operator Learning.
- The neural operator terminology is a bit loose and misused because the DeepONet, as used here is a map between finite dimensional vectors, not functions spaces.
- I believe that the sign for the 2D PDE is flipped, but perhaps this is a wrong interpretation of the derivation.

**Questions:**

- How is $p(s_i | g(z_i))$ defined?
- Can you consider non-GRF inputs, such as piece-wise, to test robustness beyond GRFs?
- How does your method compare to the baselines described above?

---

> ### Author Response · Authors · 2025-11-16
> **To Reviewer GZq4’s Report (1/2)**
>
> We thank you for your valuable comments and helpful suggestions on this work. We kindly recommend that the reviewer reads the "General response" for the details of our key contributions and the main revisions. Regarding the individual comments, we carefully consider your suggestions and address all the concerns point by point as follows.
>
> ```
> Weaknesses 1: The main novelty of the paper is the encoder engineering improvement. Different latent flow approaches such that Learning Effective Dynamics…
> ```
> **Response**: Many thanks for your constructive comments. Firstly, the core contribution of this work lies in transforming operator learning in the data space into operator learning on a lower-dimensional manifold within the latent space, and employing an OPERATOR-RNN encoder to fully leverage sparse observational data from new domains along with the sequential dynamical information it uncovers. This approach significantly enhances the robustness and accuracy of neural operator predictions (see Point 1 of the “General Response” for more details).
>
> Secondly, this paper employs a total of 8 baseline methods and 4 ablation studies, as detailed in the first paragraph of Section 4 and the first paragraph of Section 4.4. Notably, the Latent Neural ODE method has already been included as one of our baseline methods. In addition, we have selected GRUVAE and MLAE as baselines, which also fall under the category of "latent flow approaches" referenced by the reviewer. For instance, MLAE extends neural operator methods to latent spaces (published in Nature Communications in 2024) and is particularly suitable as a core baseline for this study. Furthermore, all ablation experiments are conducted within the latent space, thereby sufficiently validating the effectiveness of the proposed method. However, as you pointed out, numerous other methods in latent space of this type exist. Due to limitations in time and space, we have incorporated the several works you mentioned into the introduction section and deferred further discussion and analysis of such methods to our future work.
>
> Finally, the framework presented in this study is general and flexible, and can be extended to other neural frameworks such as diffusion models, neural ordinary differential equations, and graph neural networks. This adaptability enables more robust and accurate dynamical predictions by effectively leveraging sparse observational data in new domains.
>
>
> ```
> Weaknesses 2: DeepONet is only effective when the outputs of the neural operator lie in a linear span of trunk features, which is not the case for complex physical systems. I would suggest the author to consider the approach described in Non linear Manifold Decoders for Operator Learning.
> ```
> **Response**: Thank you for your insightful comment and helpful suggestion. We fully concur with the reviewer that this suggestion represents a highly valuable direction for improving our method. If the set of target functions lies on a nonlinear submanifold, even when the manifold itself is low-dimensional, a higher-dimensional linear space may be required to approximate it effectively. Therefore, adopting a nonlinear decoder is expected to further reduce the dimensionality $d_z$ of the latent space.
>
> We also wish to note that in many scenarios, when the Branch and Trunk networks are sufficiently deep and expressive, they can extract highly rich nonlinear features from the input functions and output coordinates. In such cases, even the subsequent simple linear dot product is often adequate to effectively couple these features, leading to excellent predictive performance in the final task. Many empirical studies across a wide range of scientific and engineering problems have demonstrated that the DeepONet architecture can achieve high prediction accuracy, underscoring the practical effectiveness and robustness for many complex systems. Therefore, our current work aims to provide a general and efficient baseline framework, while the exploration of nonlinear decoders represents a promising, forward-looking technical path to further enhance model performance or compactness in specific, more challenging contexts.
>
> We will delve deeper into this topic in future work, exploring the positive implications of nonlinear decoders on the lower bound of $d_z$ and assessing their benefit for problems where the solution manifold exhibits stronger nonlinearities.

---

> ### Author Response · Authors · 2025-11-16
> **To Reviewer GZq4’s Report (2/2)**
>
> ```
> Weaknesses 3: The neural operator terminology is a bit loose and misused because the DeepONet, as used here is a map between finite dimensional vectors, not functions spaces.
> ```
> **Response**: Many thanks for you careful reading and valuable comment. As you rightly pointed out, strictly speaking, methods such as DeepONet and FNO learn mappings between finite-dimensional vector spaces, since it is infeasible to input an infinite-dimensional function into a neural network architecture implemented on a computer. Nevertheless, a substantial body of existing work still refers to them as neural operators, primarily for the following two reasons. First, the input function $u$ is provided to DeepONet in a discretized form, meaning that when the number of sampling points is sufficiently large, most of the information about function $u$ is retained in the discretized input vector. Second, when making predictions, DeepONet can output the predicted system state at arbitrary locations, thereby achieving super-resolution in predictions and thus can be regarded as effectively outputting the solution function $s$.
>
>
> ```
> Weaknesses 4: I believe that the sign for the 2D PDE is flipped, but perhaps this is a wrong interpretation of the derivation.
> ```
> **Response**: Thank you for your valuable feedback. We have carefully re-checked the derivation of the vorticity equation. The form presented in Eq. (11), $\partial_t s=\partial_{x}\gamma \partial_{y}s - \partial_{y}\gamma \partial_{x}s + ...$, is indeed the correct form for the 2D Navier-Stokes equations. Please note that in this paper, $s$ denotes vorticity, and $\gamma$ is the stream function.
>
>
> ```
> Questions 1: How is $p(s_i|g(z_i))$ defined?
> ```
> **Response**: Many thanks for this question. In the proposed framework, $p(s_i|g_{\theta_\text{dec}}(z_i))$ is defined as a Gaussian distribution $\mathcal{N}(\mu_i, \sigma_i^2)$, where the decoder outputs both the mean $\mu_i$ and variance $\sigma_i^2$.
>
> The core reason for this parameterization is to maintain theoretical consistency with the VAE framework. It allows us to properly define the reconstruction likelihood as a probability distribution rather than a deterministic mapping. This enables: (1) Direct computation of the ELBO's reconstruction term. (2) Natural uncertainty quantification in dynamics predictions. (3) Probabilistic sampling during inference.
>
>
> ```
> Questions 2: Can you consider non-GRF inputs, such as piece-wise, to test robustness beyond GRFs?
> ```
> **Response**: Many thanks for raising this important point regarding the exploration of non-GRF inputs. We fully agree that assessing model performance on more heterogeneous and structured function spaces, such as piece-wise constant fields, is a critical step for advancing the practical robustness of neural operators. In fact, as explicitly stated in the Concluding Remarks of the original paper, we have also emphasized the significance of operator modeling under more complex distributions and identified it as a key focus for future research.
>
> In this initial work, we adopted the established GRF benchmark to provide a controlled and standardized environment. This setup allowed us to clearly isolate and demonstrate the core contribution of our robust latent framework: its superior ability to handle noise and leverage multiple sparse observations, especially through direct comparisons with state-of-the-art methods. In addition, building upon the reviewer's suggestion, our immediate next steps will include a systematic evaluation on piece-wise constant and other non-GRF inputs to further validate and enhance the model's generalization capabilities.
>
>
> ```
> Questions 3: How does your method compare to the baselines described above?
> ```
> **Response**: Many thanks for this question. Please refer to the Response to Weaknesses 1 for more details.
>
>
> Finally, we hope that our **"General response"** as well as the individual responses adequately address your concerns and reassess our work. For your convenience, the responses above are provided in a concise and direct manner. Should you have any further inquiries or require additional clarification, we look forward to your response and welcome the opportunity to engage in a more detailed discussion.

---

### Official Review · Reviewer_dJZr · 2025-10-31

**Soundness:** 3
**Presentation:** 1
**Contribution:** 2
**Rating:** 2
**Confidence:** 3

**Summary:**

The authors propose a method for learning operators between function spaces focusing on robustness to noise and ability to adapt to sparsely sampled observations. The method is named RLNO (Robust Latent Neural Operator), is designed to overcome the  degradation of that standard neural operator methods (e.g. DeepONet, FNO) when inputs are irregularly sampled or corrupted by noise.
To that end, RLNO introduces a variational autoencoder (VAE) structure around the neural operator: (1) an RNN-based encoder captures temporal and dynamical patterns from sequential or irregularly spaced observations, (2) a neural operator in latent space models the mapping between functional inputs and outputs, and (3) a decoder reconstructs the target functions in the original space.
By using a low-dimensional latent space, RLNO can tackle high-dimensional dynamical systems. Experiments on several benchmark PDE and dynamical datasets show improved accuracy and stability under noise compared to DeepONet, FNO, and related baselines.

**Strengths:**

- Aim: Paper tackles important practical problem, the robustness of NO methods to spareness of samples and noise of observations, and proposes a promising approach.
- Novelty: While each component (RNN-based temporal encoding, VAE-style probabilistic latent representation, and NO mappings) have been individually studied, RLNO’s novelty lies in integrating these three elements.
- Experiments: Baselines are appropriate and the performed empirical study shows the how RLNO overcomes the degradation of the competitors

**Weaknesses:**

First, unfortunately, I find that the paper is **not well written**. The presentation of the work is lacking on important aspects:
- Overly strong claims:  Several claims need more nuance, and some are debatable in literature,  just to name two - line 053 _"superior computational efficiency without sacrificing accuracy"_, line 268 _"thereby demonstrating the necessity and superiority of our framework design"_.
- Flow, intuition and technical complexity: Paper overly focuses to high-level presentation, while technical aspects are often introduce in an abrupt manner.  This makes the content hard to parse and key contributions hard to identify.
- Notation: For my personal taste, notations are not ideal. Further, there is inconsistencies in Eq (1) line 145 and Algorithms 2 and 3 lines 720 and 748 of the Appendix

Second, there is lack of theoretical guarantees.. claims are made citing other papers and generalising the reasoning, but not formally backed. The only try to make theoretical claim is in Theorem 1 in Appendix A.3. The claim is sloppy/not formally correct (e.g. confusing typos in lines 762, 768, domain of f is not consistent with Eq. (S6) ) and the proof is not provided but as many other thingd in the paper just briefly hinged.

Finally, since the core contribution is methodology, I find that the empirical study of two PDE problems (one 1D and other  2D) is a bit underwhelming w.r.t. publication quality requirements.

**Questions:**

NA

---

> ### Author Response · Authors · 2025-11-16
> **To Reviewer dJZr’s Report (1/3)**
>
> Many thanks for your valuable comments and helpful suggestions on this work. We kindly recommend that the reviewer reads the "**General response**" for the details of our key contributions and the main revisions. Regarding the individual comments, we carefully consider your suggestions and address all the concerns point by point as follows. We sincerely look forward to the reviewers' further evaluation of our responses and the revised version of the manuscript.
>
> ```
> Weaknesses 1: Overly strong claims: Several claims need more nuance, and some are debatable in literature, just to name two - line 053 "superior computational efficiency without sacrificing accuracy", line 268 "thereby demonstrating the necessity and superiority of our framework design".
> ```
> **Response**: Many thanks for this valuable comment. We acknowledge that the use of overly strong claims may risk appearing less rigorous in scientific research. In response, we have carefully revised the manuscript to temper our claims and provide more balanced statements, while still highlighting the key advantages of our method as supported by our experimental results.
>
> Regarding the statement about classical neural operators in the introduction: "superior computational efficiency without sacrificing accuracy." In fact, one of the core reasons neural operator methods have garnered widespread attention is their exceptionally fast prediction speed, which significantly surpasses that of traditional numerical methods, while achieving accuracy comparable to high-precision numerical approaches. To moderate the tone of expression, we have revised the phrasing to: "These advantages include significantly higher computational speed compared to conventional numerical methods while maintaining comparable predictive accuracy, as well as more effective handling of diverse inverse problems."
>
> Regarding the statement about the proposed RLNO method in the Section 4.1: "thereby demonstrating the necessity and superiority of our framework design." Here, we initially validate the advantages of our proposed method using a toy dataset of periodic orbits with unknown frequency and noise. We recognize that drawing definitive conclusions based solely on this toy dataset may be too strong; indeed, such conclusions should be supported by experiments on four more complex PDE systems along with various ablation studies. Therefore, we have revised this part to state: “thereby preliminarily demonstrating its advantages in terms of noise robustness and effective utilization of sparse observational data.”
>
> Finally, we have conducted a thorough review of the entire text, tempering or removing overly strong claims to thereby enhance the paper’s overall readability.
>
>
> ```
> Weaknesses 2: Flow, intuition and technical complexity: Paper overly focuses to high-level presentation, while technical aspects are often introduced in an abrupt manner. This makes the content hard to parse and key contributions hard to identify.
> ```
> **Response**: Thank you for your constructive feedback. We fully acknowledge that readability is crucial for the effective dissemination of technical content in academic papers, and we strive to present our work in a manner that is both concise and clear. Firstly, to facilitate a more comprehensive understanding of our core contributions, we provide a detailed elaboration in the Point 1 of the “General Response”.
>
> Secondly, we acknowledge that the RLNO framework presented in this paper, which involves a synergistic integration of techniques including neural operators, variational autoencoders, and latent spaces, may pose challenges in comprehension. Therefore, readers are encouraged to refer to the descriptions in the Methods section to better comprehend the methodological framework illustrated in Figure 1, thereby enabling a clearer understanding of the core methodology presented in this paper. In addition, we also provide the pseudocode for the training and testing procedures of the RLNO method in Appendix A.2 to help readers develop a clear understanding of the execution flow.
>
> Finally, we have thoroughly reviewed the entire manuscript, striving to minimize high-level presentation while enriching the technical details with reference to the framework illustrated in Figure 1. These revisions aim to enhance the overall readability and clarity of the paper, allowing readers to better comprehend our methodology and core contributions (key modifications are succinctly listed in Point 2 of the "General Response").

---

> ### Author Response · Authors · 2025-11-16
> **To Reviewer dJZr’s Report (2/3)**
>
> ```
> Weaknesses 3: Notation: For my personal taste, notations are not ideal. Further, there is inconsistencies in Eq (1) line 145 and Algorithms 2 and 3 lines 720 and 748 of the Appendix.
> ```
> **Response**: Many thanks for your careful reading and helpful advice. Given that our method involves multiple components—such as data space, latent space, an OPERATOR-RNN encoder, a latent neural operator, and a decoder—the notation in this paper may appear relatively extensive. Therefore, to help readers quickly clarify the meaning and relationships of the notations, our framework diagram in the main text (see Figure 1) includes most of the core notations, while a summary of notations is supplemented in Appendix A.2 (see Table S1).
>
> In addition, the notation in Equation (1) is, in fact, consistent with the subsequent text, where $z_i$ denotes the system state at the $i$-th step in the latent space, and $s_i$ represents the system state at the $i$-th step in the data space; $p(z_0)$ is the distribution of the initial value $z_0$ in latent space, assumed to be Gaussian, with its mean and variance estimated from sparse observations via the OPERATOR-RNN encoder; OPERATORsolve denotes the process of performing operator computations; $u$ is the input function to the neural operator, such as initial/boundary conditions or parametric functions; $g_{\theta_\text{dec}}$ denotes the decoder, and $\tilde{g}_{\theta_\text{enc}}$ denotes the encoder.
>
> Finally, to ensure the rigor of this paper, we have carefully verified all notation used in this work, and included the key notation in the Table S1 in Appendix A.2.
>
>
> ```
> Weaknesses 4: Second, there is lack of theoretical guarantees.. claims are made citing other papers and generalising the reasoning, but not formally backed...
> ```
> **Response**: Many thanks for your careful reading and constructive feedback. My sincere apologies for any confusion caused by the earlier typos and the inadvertent error. Firstly, the continuous vector-valued function should be $\bf{f}: \mathbb{R}^{l+d_z}\to \mathbb{R}^m$, where $d_z$ denotes the dimensionality of the latent space. In addition, in Line 762 of the original text, it should be corrected to $K_1 \subset \mathcal{X}$, $K_2 \subset \mathbb{R}^d$; and in Line 768, $\langle \cdot,\cdot \rangle$ should refer to the inner product in $\mathbb{R}^m$.
>
> Secondly, as you noted, the core contribution of this work lies in the methodology, with a greater emphasis on building a scalable machine learning framework and conducting experimental analyses across multiple systems. As a deep learning approach, theoretical analysis remains a significant challenge in current research, and we intend to explore this further in future studies. Nevertheless, we provide a theoretical support for the proposed method through Theorem 1. To ensure rigor, we have carefully examined the theorem and its proof to guarantee their correctness.
>
> Thirdly, regarding the proof of Theorem 1, we should clarify that the proof we provide is brief, this is because it builds directly upon Theorem 2 in Reference [Lu et al. 2021]. Notably, in proving the universal approximation theorem for the classical DeepONet approach (i.e., Theorem 2 in Reference [Lu et al. 2021]), Lu et al. also selected specific values for the branch and trunk network, reducing it to the universal approximation theorem for operators proposed by Chen et al. (Reference [Chen et al. 1995]), and provided a very concise proof by leveraging the universal approximation theorem of neural networks. Similarly, here, by selecting specific values for $\bf{f}$ and $\tilde{\bf{f}}$ in Theorem 1, we reduce it to Theorem 2 in Reference [Lu et al. 2021], thus establishing a concise proof. Furthermore, the branch network of the proposed RLNO incorporates the latent value $z_0$ estimated by the OPERATOR-RNN encoder (see Eq. S7 in the Appendix), thereby additionally integrating sparse observation states and extracted sequential dynamics information. This design contributes to the superior performance of our method compared to baseline approaches in the experimental analysis.
>
> Finally, as a promising extension of neural operator methods, we believe that much of the existing theoretical work on classical neural operators can be extended to our RLNO framework. Given that this is not the core contribution of this paper, more detailed theoretical analysis and discussion will be addressed in future work.
>
> [1] Lu Lu, Pengzhan Jin, Guofei Pang, Zhongqiang Zhang, and George Em Karniadakis. Learning nonlinear operators via deeponet based on the universal approximation theorem of operators. Nature machine intelligence, 3(3): 218–229, 2021.
>
> [2] T. Chen and H. Chen. Universal approximation to nonlinear operators by neural networks with arbitrary activation functions and its application to dynamical systems. IEEE Transactions on Neural Networks, 6(4):911917, 1995.

---

> ### Author Response · Authors · 2025-11-16
> **To Reviewer dJZr’s Report (3/3)**
>
> ```
> Weaknesses 5: Finally, since the core contribution is methodology, I find that the empirical study of two PDE problems (one 1D and other 2D) is a bit underwhelming w.r.t. publication quality requirements.
> ```
> **Response**: Thank you for your valuable comment. In fact, in this work, we select a total of five systems for experimental validation, including a toy dataset with periodic orbits of different frequencies; two 1D PDE systems: the Diffusion-Reaction (DR) equation and the Kuramoto–Sivashinsky (KS) equation; as well as two 2D PDE systems: the Navier–Stokes equations and Rayleigh–Bénard (RB) convection. Several of these PDE systems exhibit complex chaotic behavior, making them ideal testbeds for validating neural operator methods. In addition, we also conduct extensive ablation studies and robustness analysis in Section 4.4, thereby validating the effectiveness of the proposed method.
>
>
> Finally, we hope that our **"General response"** as well as the individual responses adequately address your concerns and reassess our work. For your convenience, the responses above are provided in a concise and direct manner. Should you have any further inquiries or require additional clarification, we look forward to your response and welcome the opportunity to engage in a more detailed discussion.

---

> > ### Comment · Reviewer_dJZr · 2025-11-26
> >
> > I thank the authors for their replies. Since some of my concerns have been appropriatly adressed, I rise my score.

---

> ### Author Response · Authors · 2025-11-27
>
> Thank you once again for raising your score and for your constructive feedback—we truly appreciate your recognition of our efforts in addressing some of the initial concerns.
>
> We believe the work presented in this paper holds strong practical potential and extensibility, as further elaborated in our **General Response**. Our approach not only enables effective utilization of multiple sparse observational datasets, but can also be naturally extended to other neural frameworks—such as **GenCast for weather forecasting [1]** and beyond. To give a concrete example, the method presented in [1] could leverage the ideas introduced in our paper to incorporate a wider range of observational data, thereby further improving prediction accuracy. Similar extensions can be broadly applied to many other related studies as well.
>
> If there are any remaining questions or aspects that you feel require further clarification, we would be glad to provide additional details or engage in further discussion during the remaining rebuttal period.
>
> We sincerely value your time and insights.
>
> [1] Price I, Sanchez-Gonzalez A, Alet F, et al. Probabilistic weather forecasting with machine learning[J]. Nature, 2025, 637(8044): 84-90.

---

### Official Review · Reviewer_UyLw · 2025-10-31

**Soundness:** 3
**Presentation:** 3
**Contribution:** 3
**Rating:** 6
**Confidence:** 4

**Summary:**

The work builds upon the development of neural operators, which has been a highly cited and used method for learning maps between measurements to full state estimates. Or more broadly mapping functions to functions.

The key idea in this paper is to do the neural operator mapping in latent space.  That is the basic innovation of the paper, and the authors show that his is a much more way to learn the operator than in the original measurement space.  This is consistent across many emerging real-world examples:  work in the latent space instead of the measurement space in order to improve performance.

**Strengths:**

This is a very solid and useful contribution to the field of neural operators.   A clear and important in the next step of neural operators as exploiting the latent space is clearly what should be done.  The results back this up.

**Weaknesses:**

Not many examples were presented, and some are not very convincing (equation 11 is linear PDE is it not?).  I think something like 2D Kolmogorov flow in the turbulent regime would be a much more convincing set of data than what they have.  So I found the examples not to the level of where they should be.

**Questions:**

It seems the "sparsity" has not been well characterized in how this works?  Do the authors have a metric for this?  It is important to establish when the sampling will actually work or not.

How long can the roll out in latent space go?  Most latent space long-time roll outs eventually diverge or break.  Is there any guarantee about a stable long-term roll out?

---

> ### Author Response · Authors · 2025-11-16
> **To Reviewer UyLw’s Report (1/2)**
>
> We would like to thank the reviewer for the overall positive feedback and helpful suggestions. We also kindly recommend that the reviewer reads the "**General response**" for a comprehensive overview of our key contributions and the main revisions.
>
> ```
> Weakness: Not many examples were presented, and some are not very convincing (equation 11 is linear PDE is it not?). I think something like ...
> ```
> **Response**: Thank you for your valuable comments and helpful suggestions. In this work, we have selected a total of five systems for experimental validation, including a toy dataset with periodic orbits of different frequencies; two 1D PDE systems: the Diffusion-Reaction (DR) equation and the Kuramoto–Sivashinsky (KS) equation; as well as two 2D PDE systems governed by the Navier–Stokes equations and Rayleigh–Bénard (RB) convection.
>
> Regarding the NS system (Equation (11)), we would like to clarify that it constitutes a **nonlinear coupled PDE system**. The nonlinearity arises explicitly from the advection-like terms $\partial_{x}\gamma \partial_{y}s - \partial_{y}\gamma \partial_{x}s$. While the diffusion term is linear, the coupling between the field $s$ and the stream function $\gamma$ via the Poisson equation $\Delta \gamma = -s$ makes the overall system nonlinear. This form is analogous to the vorticity-stream function formulation used in fluid dynamics and exhibits nontrivial, advection-dominated dynamics.
>
> Furthermore, the **Rayleigh–Bénard (RB) convection system** (governed by the dynamical equations in Appendix E) represents a canonical and highly challenging turbulent flow. It includes coupled nonlinear PDEs for momentum and heat transport, exhibiting rich spatiotemporal chaos and buoyancy-driven turbulence. Together with the 2D Navier-Stokes equation, the RB system offers a strong test case for complex multi-physics phenomena.
>
> Therefore, we appreciate the reviewer's suggestion of the 2D Kolmogorov flow. If time permits, we will include it as an additional turbulent example to further strengthen our validation in the future. In addition, to prevent potential misunderstandings, we have revised the manuscript to better emphasize the nonlinear nature of the NS system and the complex and challenging characteristics of the RB system.

---

> ### Author Response · Authors · 2025-11-16
> **To Reviewer UyLw’s Report (2/2)**
>
> ```
> Question 1: It seems the "sparsity" has not been well characterized in how this works? Do the authors have a metric for this? It is important to establish when the sampling will actually work or not.
> ```
> **Response**: Many thanks for your careful reading and valuable comments. In fact, the sparsity here refers to the scarce and irregularly sampled observational data in complex systems. Specifically, when a system operates in a new domain, due to observational costs or inherent system complexity, we can only obtain state measurements at a limited number of time instances (denoted as $T_{\text{enc}}$), which may also be heavily contaminated by noise. This poses significant challenges for existing methods such as classical neural operators. In contrast, the trained RLNO approach can effectively leverage these sparse observations, uncovering underlying dynamical information embedded within the sequential data, and encode it into a latent representation, thereby enabling more accurate operator learning and dynamical forecasting.
>
> Furthermore, the metric for this sparsity observation is jointly determined by two parameters, $\lambda$ and $T_\text{enc}$. Specifically, in our simulation experiments, we first generate equally spaced observational data from the dynamical equations, then randomly retain a proportion $\lambda$ of the observed samples, and finally take the first $T_\text{enc}$ samples as the sparse observations in the new domain. This setup is, in fact, consistent with many real-world scenarios. Under this configuration, we also conduct an ablation study in Fig. 5c to investigate the effects of varying $T_\text{enc}$. The experimental results demonstrate that our method achieves the best performance across different sparsity levels. Moreover, as $T_\text{enc}$ increases, the OPERATOR-RNN encoder is able to extract more useful dynamical information, leading to more accurate operator learning.
>
> ```
> Question 2: How long can the roll out in latent space go? Most latent space long-time roll outs eventually diverge or break. Is there any guarantee about a stable long-term roll out?
> ```
> **Response**: Many thanks for your insightful comments. Notably, as a classic deep learning approach, neural operators require sufficient training data to effectively learn a family of dynamical systems. Once trained, the neural operator method enables extremely fast solutions in new domains while maintaining accuracy, significantly outperforming conventional numerical solvers. Consequently, our experimental findings suggest that, given sufficient training data, the latent space can effectively capture the lower-dimensional manifold underlying the data space. By learning the dynamics on this manifold within the latent space, the approach not only preserves the complexity of dynamic processes in the original data space but also reduces the complexity of the operator learning task. This approach consequently enables more accurate long-term predictions (corresponding to larger values of $T$). In addition, from a theoretical perspective, we also provide a universal approximation theorem for RLNO in Appendix A.3, along with a streamlined proof.
>
> In addition, it should be noted that neural operator methods circumvent the need for integral or numerous iterative computational procedures. Consequently, they mitigate the issue of rapid error accumulation in certain stiff or chaotic systems, often resulting in more stable performance. The proposed RLNO method, as a member of the neural operator family, inherently inherits this advantageous characteristic.
>
>
> Finally, we hope that our **"General response"** as well as the individual responses adequately address your concerns. For your convenience, the responses above are provided in a concise and direct manner. Should you have any further inquiries or require additional clarification, we look forward to your response and welcome the opportunity to engage in a more detailed discussion.

---

### Author Response · Authors · 2025-11-16
**General response:**

We would like to thank the reviewers for your time, efforts, and valuable comments and suggestions, which do help us to significantly improve the quality of this work. Here, we succinctly summarize the novelty and contributions of this work and then list the main revisions implemented in the manuscript.

**Point 1: the novelty and contributions of this work**

(1) A Novel Fusion Framework. RLNO represents a novel approach grounded in VAE framework, incorporating an RNN-based encoder, latent neural operator and a decoder. This framework effectively harnesses the strengths of RNNs and neural operators, enabling more accurate operator learning. Specifically, the core idea of the proposed framework is to uncover the underlying low-dimensional manifold structure in observed data through latent space transformations. On this basis, the dynamical laws on this manifold can be learned in the latent space, and subsequently operator modeling can be achieved through a decoder. Experimental results demonstrate that this approach can more efficiently model a family of dynamical systems and achieve operator learning.

(2) Utilizing More Dynamic Information. The proposed OPERATOR-RNN encoder extracts and leverages more domain-specific information and dynamic evolution patterns from the sequential observational data in new domains. It remains applicable under non-uniform sampling conditions, thereby improving the robustness and accuracy of the RLNO method. Specifically, current neural operator methods, including DeepONet and FNO, often only input the sampled functions (such as parameter functions, initial or boundary conditions) when tested in new domains. However, in many practical applications, there may exist prior observational data at several moments in this new domain, which is often overlooked in previous methods. Our research finds that fully leveraging these prior observational states has a very significant effect on promoting the robustness and accuracy of neural operator modeling. This benefit arises from domain-specific observations, which not only provide more state information but also extract specific dynamic information from these sequential observations.

(3) Efficiency and Scalability. RLNO inherits the low computational costs of neural operators, significantly outperforming RNN and Neural ODE-based methods. And it can select a smaller latent space dimension, which reduces learning complexity and data requirements, enabling easier extension to high-dimensional complex systems. In addition, the framework presented in this study is general and flexible, and can be extended to other neural frameworks such as diffusion models, neural ordinary differential equations, and graph neural networks. This adaptability enables more robust and accurate dynamical predictions by effectively leveraging sparse observational data in new domains.

Finally, we select a total of five systems for experimental validation, encompassing a synthetic dataset with periodic orbits of varying frequencies; two 1D PDE systems: the Diffusion-Reaction equation and the Kuramoto–Sivashinsky equation; as well as two 2D PDE systems: the Navier–Stokes equations and Rayleigh–Bénard convection. Additionally, we select 8 baseline methods (including approaches based on RNNs, classical neural ODEs, classical neural operators and latent neural operators), and conducted 4 ablation studies to affirm the significant contribution of each component in our methodology. To enhance the reliability and reproducibility of our experimental results, we also provide all the experiment codes in the supplementary materials.


**Point 2: the main revisions implemented in the manuscript**

(1) We supplement the Key Notation table in Appendix A.2 to enhance the clarity and readability of the article.

(2) We meticulously review the entire text and tone down overly strong claims. Examples include the conclusion of the second paragraph in the Introduction section and the second paragraph of Section 4.1.

(3) We refine the statement of Theorem 1 and carefully verify its proof in Appendix A.3. Furthermore, we supplement the corresponding discussions following the proof.

(4) In Section 4.3, we further elaborate on the complex nature of the 2D Navier–Stokes and Rayleigh–Bénard PDE systems, thereby demonstrating that the selected experimental systems can be used to validate the effectiveness of the proposed method.

(5) We supplement several references on “latent flow approaches” in the introduction section.

(6) We carefully review the technical details in the paper to enhance readability. This includes supplementing and refining the notation explanations for Eqs. (2) and (10), and correcting the typographical errors in Line 289 of the original text.

Finally, we thank all the reviewers again for your valuable and insightful comments. We hope that our General Response as well as the individual responses for each reviewer adequately addresses the reviewers’ concerns.

---

### Note · Authors · 2026-07-20

I have read and agree with the venue's withdrawal policy on behalf of myself and my co-authors.

---

### Meta-Review · Area_Chair_ETX1 · 2026-01-05

**Summary:**

While the authors addressed several presentation and clarification issues and improved parts of the rebuttal, a number of substantive concerns remain unresolved. In particular, reviewers raised persistent questions regarding novelty, adequacy of baselines relative to recent neural operator methods, and whether the proposed framework fully satisfies the definition of a neural operator. The paper received mixed evaluations, with reviewer scores ranging from positive to negative and an overall average rating of approximately 5.5. Notably, the highest score of 8 was given by Reviewer KSwG with low confidence 2, and this review did not articulate substantial technical strengths that would outweigh the remaining concerns. Given these factors, the overall assessment falls below the acceptance threshold.

**Reviewer Concerns:**

Reviewer UyLw

- Concerns that are potentially addressed:

    - Definition of sparsity: The authors clarified what is meant by sparsity, how sparse observations are generated, and how sparsity is controlled through experimental parameters, which reasonably addresses this concern.

- Concerns that might not be addressed:

     - Limited number of benchmark examples: Although the authors emphasized the relevance of the selected systems, they did not introduce new or more challenging benchmarks, which makes it unlikely that this concern would lead to a score increase.

     - Long rollout behavior: The authors provided general arguments about stability and universal approximation, but they did not directly answer the reviewer’s question regarding guarantees or empirical characterization of long term rollouts in latent space at a level that would likely affect the reviewer’s score.


Reviewer dJZr

- Concerns that are potentially addressed:

    - Writing and presentation: The authors revised the manuscript to tone down overly strong claims, improve flow, and clarify notation, which partially addresses the reviewer’s presentation concerns.

- Concerns that might not be addressed:

    - Lack of theoretical guarantees: Although the authors referenced a theorem in the appendix, the reviewer’s concern regarding insufficient or weak theoretical grounding remains largely unresolved.

    - Limited number of benchmark problems: The empirical evaluation remains relatively narrow given the methodological claims, and this concern is unlikely to be fully addressed by the current revisions.


Reviewer GZq4

- Concerns that are potentially addressed:

    - Correctness of the 2D PDE formulation: The authors clarified the nonlinear nature of the Navier Stokes and Rayleigh Benard systems, which addresses this concern.

- Concerns that might not be addressed:

    - Novelty: While the application context may be new, the proposed method largely combines existing components, namely latent space modeling and neural operators, by replacing latent neural ODE dynamics with DeepONet style operators. This limits the perceived novelty.

    - Baselines: Although multiple baselines are included, they appear outdated and omit several recent neural operator approaches. If the paper positions itself as proposing a new neural operator framework, stronger and more recent neural operator baselines would be expected.

    - Function space mapping and operator definition: The proposed framework relies on fully discrete RNN cells in the encoder, which are not resolution invariant. This raises a legitimate question about whether the method fully qualifies as a neural operator in the strict sense, and this concern is not convincingly resolved by the authors’ response.

Reviewer KSwG

- Concerns that are potentially addressed:

    - Presentation issues: Typos and minor clarity issues were corrected.

- Concerns that might not be addressed:

     - Computational timing and efficiency: The authors provided qualitative statements about efficiency but did not include concrete measurements such as wall clock time or runtime comparisons, which could have directly addressed this concern.


Common concerns across the reviewers:

- The novelty of the proposed method is limited, as it primarily combines existing components such as latent modeling and neural operators without a clearly distinct conceptual or methodological advance.

- The set of baselines is insufficient relative to recent neural operator literature, and stronger or more up-to-date neural operator comparisons are missing.

- It is unclear whether the proposed framework fully qualifies as a neural operator, given its reliance on discrete RNN components that are not resolution invariant.

- The empirical evaluation is limited in scope, with relatively few benchmark problems compared to the breadth of the methodological claims.

**Reviewer Scores:**

Reviewer UyLw: 6 with confidence 4
Reviewer dJZr: updated to 4 with confidence 3
Reviewer GZq4: 4 with confidence 3
Reviewer KSwG: 8 with confidence 2

The average score is approximately 5.5. Given the halted discussion, further score changes appear unlikely. Considering that the highest score was given with low confidence and limited articulation of strengths (8 with confidence 2 --- I would not expect that the reviewer would champion the paper) and that significant concerns remain regarding baselines, novelty, and operator definition, the overall evaluation falls below the acceptance threshold.

---

### Decision · Program_Chairs · 2026-01-26

Reject